



**Characterization of submicron particles by Time-of-Flight Aerosol Chemical Speciation Monitor (ToF-ACSM) during wintertime: aerosol composition, sources and chemical processes in Guangzhou, China**

Junchen Guo[1], Shengzhen Zhou[1,8], Minfu Cai[1], Jun Zhao[1,8], Wei Song[2], Weixiong Zhao[3], Weiwei Hu[2], Yele Sun[4], Yao He[4], Chengqiang Yang[3], Xuezhe Xu[3], Zhisheng Zhang[5], Peng Cheng[6], Qi Fan[1], Jian Hang[1], Shaojia Fan[1], Xinming Wang[2], Xuemei Wang[7]

[1]School of Atmospheric Sciences, Guangdong Province Key Laboratory for Climate Change and Natural Disaster Studies, and Institute of Earth Climate and Environment System, Sun Yat-sen University, Guangzhou, 510275, P. R. China

[2]State Key Laboratory of Organic Geochemistry, Guangzhou Institute of Geochemistry, Chinese Academy of Sciences, Guangzhou, 510640, P. R. China

[3]Laboratory of Atmospheric Physico-Chemistry, Anhui Institute of Optics and Fine Mechanics, Chinese Academy of Sciences, Hefei, 230031, P. R. China

[4]State Key Laboratory of Atmospheric Boundary Layer Physics and Atmospheric Chemistry, Institute of Atmospheric Physics, Chinese Academy of Sciences, Beijing 100029, P. R. China

[5]South China Institute of Environmental Sciences, Ministry of Ecology and Environment, Guangzhou, 510655, China

[6] Institute of Technology on Atmospheric Environmental Safety and Pollution Control, Jinan University, Guangzhou, 510632, P. R. China

[7]Institute for Environmental and Climate Research, Jinan University, Guangzhou, 511443, P. R. China

[8]Southern Marine Science and Engineering Guangdong Laboratory (Zhuhai), Zhuhai, 519082, P. R. China

*Correspondence to:* Shengzhen Zhou (zhoushzh3@mail.sysu.edu.cn); Jun Zhao (zhaojun23@mail.sysu.edu.cn)

**Abstract**. Particulate matter (PM) pollution in China is an emerging environmental issue which policy makers and public have increasingly paid attention to. In order to investigate the characteristics, sources, and chemical processes of PM pollution in Guangzhou, a field measurement was conducted from 20 November 2017 to 5 January 2018, with a Time-of-Flight Aerosol Chemical Speciation Monitor (ToF-ACSM) and other collocated instruments. Mass concentrations of non-refractory submicron



particulate matters (NR-PM$_1$) measured by the ToF-ACSM were correlated well with those of PM$_{2.5}$ or

PM$_{1.1}$ measured by filter-based methods. The organic mass fraction increased from 45% to 53% when the air switched from non-pollution periods to pollution episodes, indicating significant roles of organic aerosols (OA) during the whole study. Based on the mass spectra measured by the TOF-ACSM, Positive Matrix Factorization (PMF) with multilinear engine (ME-2) algorithm was performed to deconvolve OA into four factors, including hydrocarbon-like OA (HOA, 12%), cooking OA (COA,

18%), semi-volatile oxygenated OA (SVOOA, 30%), and low-volatility oxygenated OA (LVOOA, 40%). Furthermore, we found that SVOOA and nitrate were significantly contributed from local traffic emissions while sulfate and LVOOA were mostly attributed to regional pollutants. Comparisons between this work and other previous studies in China show that SOA fraction in total OA increases spatially across China from the North to the South.

Two distinctly opposite trends for NR-PM$_1$ formation were observed during non-pollution period and pollution EPs. The ratio of secondary PM (SPM = SVOOA + LVOOA + sulfate + nitrate + ammonium) to primary PM (PPM = HOA + COA + chloride), together with peroxy radicals RO$_2$* and ozone, increased with increasing NR-PM$_1$ concentration during non-pollution period, while an opposite trend of these three quantities was observed during pollution EPs. Furthermore, oxidation degrees of

both OA and SOA were investigated using the $f_{44}/f_{43}$ space and the results show that at least two OOA factors are needed to cover a large range of $f_{44}$ and $f_{43}$ in Guangzhou. Comparisons between our results and other laboratory studies imply that volatile organic compounds (VOCs) from traffic emissions in particular from diesel combustion and aromatic compounds are most possible SOA precursors in Guangzhou. Peroxy radical RO$_2$* was used as a tracer for SOA formed through gas phase oxidation. For

non-pollution period, SOA concentration was reasonably correlated with RO$_2$* concentration during both daytime and nighttime, suggesting that gas phase oxidation was primarily responsible for SOA formation. For pollution EPs, when NR-PM$_1$ mass concentration was divided into six segments, in each segment except for the lowest one, SOA concentration was correlated moderately with RO$_2$* concentration, suggesting that gas phase oxidation still plays important roles in SOA formation. In

addition, the slopes of linear regressions for the above correlations increased with increasing NR-PM$_1$ mass concentration, probably representing enhanced gas-to-particle partitioning under high NR-PM$_1$ concentration. The intercepts of the above linear regressions, which correspond to the extent of other mechanisms (i.e., heterogeneous and multiphase reactions), increased with increasing NR-PM$_1$ mass concentration. Our results suggest that while gas phase oxidation contributes predominantly to SOA





formation during non-pollution periods, other mechanisms such as heterogeneous and multiphase reactions play more important roles in SOA formation during pollution EPs than gas phase oxidation.

## 1 Introduction

With rapid development of human civilization, more attention is paid to air quality by public,
government, and scientists, especially in developing countries like China. In recent years, particulate matter (PM) pollution has become one of the most concerned environmental issues because of its significant effects on both climate change (IPCC 2013) and human health (Pope III et al., 2006). Atmospheric aerosols exert radiative forcing directly through scattering or absorbing solar radiation or indirectly through cloud formation. In addition, previous studies in the last decade have shown that
respiratory and cardiovascular diseases are highly related to fine particles, revealing significant deleterious effects of ambient aerosols on human health (Kreyling et al., 2006). Thus, knowledge of chemical composition, formation mechanisms, and potential sources of fine particles is essential for both academic community and the public, since it is still currently very limited even through decades of investigation. In last decades, most studies focused on $PM_{2.5}$ and made significant progresses while less
attention was paid to submicron particles (i.e., $PM_1$). For instance, although China National Ambient Air Quality Standard (CNAAQS) for $PM_{2.5}$ was established in 2012, the corresponding national standard for submicron particles such as $PM_1$ has not yet been set up. In fact, it has been shown that $PM_1$ particles may cause much more damage to human health than $PM_{2.5}$ due to their smaller sizes which make them more easily access to human bodies (Ibald-Mulli et al., 2002; Kreyling et al., 2006).
Therefore, more extensive and in-depth studies should be conducted for submicron particles besides $PM_{2.5}$ to obtain a comprehensive understanding on health and climate impacts of fine particles.

Field measurements of aerosol chemical composition mainly employ filter-based offline and mass spectrometric online technologies. Although traditional filter-based methods which are still widely used contribute substantially to understanding of bulk aerosol chemical composition, its obvious
shortcomings which include low time resolution from hours to days and evaporative loss limit the capacity of this technology in aerosol measurements. In comparison, mass spectrometric online methods have higher time resolutions varying from seconds to hours, therefore proven to be an efficient way for measurements of aerosol mass concentration and chemical composition (Aiken et al., 2009; Zhang et al., 2011; Sun et al., 2013; Crippa et al., 2013; Sun et al., 2014; Lee et al., 2015; Hu et al., 2017). For
example, aerosol mass spectrometer (AMS) is one such instrument that is widely employed in aerosol





chemical composition measurements for its reliable data quality and relatively high mass resolution. However, the full version of AMS tends to be costly and time-consuming in terms of its operation and maintenance. As a simplified version of AMS, Aerosol Chemical Speciation Monitor (ACSM) has been widely adopted in recent years among research institutions and environmental monitoring stations for its

relatively simple operation, robustness, low cost and sufficient time resolution for field observations spanning months or longer (Allan et al., 2010; Ng et al., 2011; Sun et al., 2013; Fröhlich et al., 2013; Sun et al., 2014; Sun et al., 2016). Certainly, this simplified design is evitable to bring a few disadvantages for ACSM. Compared with other types of AMS, ACSM gives up the ability of measuring particle size distribution, which makes users lose a robust tool to characterize ambient particulate

matters and identify potential sources (Frank et al., 2005; Ge et al., 2012; Lee et al., 2017). Besides, limited by relatively poorer resolution, mass spectra collected by ACSM cannot execute high-resolution peak fitting (Timonen et al., 2016). This incapability prevents users from obtaining some further information such as elemental ratio in particulate matters, which is essential to our knowledge about climate effects or toxicity of aerosols (Aiken et al., 2007; Chhabra et al., 2011; Canagaratna et al., 2015).

The simplification of fragmentation table for ACSM, based on ambient AMS data, also likely leads to some small deviations and makes this instrument unsuitable for laboratory studies requiring high precision (Jimenez et al., 2003; Allen et al., 2004; De Haan et al., 2009).

Numerous studies were conducted to investigate chemical composition, sources and secondary processes of PM pollution through AMS or ACSM. It has been shown that organics generally account

for a large proportion of PM (Zhang et al., 2007; Aiken et al., 2009; Allan et al., 2010; DeCarlo et al., 2010; Sun et al., 2013; Lee et al., 2015; Bressi et al., 2016; Li et al. 2017). In previous studies, mass spectral signals were input into the Positive Matrix Factorization (PMF) to explore source information of organic aerosols (Zhang et al., 2011). Generally, OA can be deconvolved into primary organic aerosols (POA) which can be further classified according to different markers of primary emissions

(e.g., HOA, COA and BBOA), and oxygenated organic aerosols (OOA) which can be further resolved based on oxidation degree (e.g., SVOOA and LVOOA). However, detailed source features are diverse at different regions. For example, Aiken et al. (2009) identified industry-induced local nitrogen-containing OA (LOA), which was barely reported in other cities such as Beijing (Sun et al., 2018), London (Allan et al., 2010) or Paris (Crippa et al., 2013). Similarly, biomass burning OA

(BBOA) which was clearly identified at some sites such as those in Hebei (Huang et al., 2019), Mexico (Aiken et al., 2009) or Fresno (Ge et al., 2012), was missing in other locations (during same seasons)





such as Hong Kong (Sun et al., 2016), or Beijing (Sun et al., 2013). Although OA sources have obvious spatial distinction, Jimenez et al. (2009) found a common trend that the fraction of OOA (especially LVOOA) increases from urban to rural areas.

Physical/chemical characteristics and secondary processes of PM were investigated in previous studies (Liggio et al., 2006; Shilling et al., 2009; Marais et al., 2016). Photochemistry which highly depends on daytime solar radiation is commonly believed to play a dominant role in formation of secondary PM (SPM = ammonium + nitrate + sulfate + OOA); However, recent studies showed that the contribution from aqueous reactions or heterogeneous reactions cannot be ignored. For instances, it has

been found that the heterogeneous hydrolysis of $N_2O_5$ on the surface of deliquescent aerosols is a significant pathway for nitrate formation during nighttime (Wang et al., 2018; Wen et al., 2018). Sun et al. (2013) proposed that the mass concentration of sulfate substantially increased through fog processes. Besides, numerous studies have suggested that the amount of SOA from reactive uptake of water-soluble VOCs is comparable to that from gas phase oxidation (Ervens et al., 2011; McNeill, 2015;

Herrmann et al., 2015; Marais et al., 2016).

        Results from previous studies suggested that primary emissions, secondary chemical processes, meteorological conditions and regional transport are possible major factors that determine local PM concentration but significantly vary with seasons and locations (Sun et al., 2013; Lee et al., 2015; Sun et al., 2016; Bressi et al., 2016; Gani et al., 2019). Therefore, it is essential to conduct continuous field

measurements at various locations to investigate the PM formation mechanisms, as well as the temporal and spatial evolution. Guangzhou, a highly developed city in the Pearl River Delta (PRD) region, is considered to be one of the most densely populated cities in China. However, previous studies on local PM characteristics in Guangzhou were mainly conducted by filter-based methods with low time resolutions (Tan et al., 2009; Zhang et al., 2010; Tao et al., 2012) and therefore knowledge of detailed

PM characteristics is still lacking. It is hence urged to perform field measurements with high-time resolutions in the city.

        In this study, we employed Time-of-Flight Aerosol Chemical Speciation Monitor (ToF-ACSM) to measure chemical composition and mass concentrations of submicron particles at an urban site in Guangzhou which locates at Guangzhou Institute of Geochemistry, Chinese Academy of Sciences. The

following sections present measurement techniques for gases and aerosols, followed by the methodology for data analyses. In the results and discussion section, temporal variations of chemical composition and mass concentration for NR-PM$_1$ are illustrated, followed by the source apportionment



for OA and the diurnal profiles for NR-PM$_1$. The implications of possible mechanisms responsible for wintertime secondary organic aerosol formation in the subtropical urban areas are discussed.

## 2 Experimental

### 2.1 Measurement site and data

Chemical compositions of NR-PM$_1$ consisting of Cl$^-$, SO$_4^{2-}$, NO$_3^-$, NH$_4^+$, and organics were measured from 20 November 2017 to 5 January 2018 by a ToF-ACSM at an urban air quality
monitoring site locating at Guangzhou Institute of Geochemistry, Chinese Academy of Sciences (GIG). Detailed descriptions of ToF-ACSM can be found elsewhere (Fröhlich et al., 2013) and the methodology for data analysis is presented in the next section. The site is surrounded by three major traffic roads including Guangyuan road, Huanan road and Keyun road. Several university campuses are located to north and west of the site while commercial buildings, restaurants, and residential areas are
next to its east and south. In addition to aerosol measurements by the ToF-ACSM, ambient gas species such as NO$_x$ and O$_3$ were measured at the same site by various gas analyzers (Thermo Scientific, USA) while filter samples were collected by an Anderson nine-stage sampler. Meteorological data (wind speed and direction, relative humidity, temperature and pressure) and PM$_{2.5}$ concentrations were measured about 2 and 4 kilometers away from sample site by South China Institute of Environmental
Science and Guangzhou Environmental Monitoring Center respectively. Total peroxy radicals (RO$_2$* = ΣRO$_2^\bullet$ + HO$_2$) were measured by the peroxy radical chemical amplification technique at the same sampling site. The measurement technique and detailed results could be found in Yang et al. (2018, 2019).

### 2.2 ACSM data analysis

The final mass concentrations and mass spectra were processed from the raw ToF-ACSM data by a standard ACSM data analysis software (Tofware, version 2.5.13) based on Igor pro (version 6.37), with the widely applied procedures described in Ng et al. (2011b) and Sun et al. (2012). Based on the on-site calibrations, a relative ionization efficiency (RIE) value of 1.2 and 3.3 was obtained for sulfate and ammonium respectively. A RIE value of 1.1, 1.3 and 1.4 was respectively adopted for nitrate, chloride
and organics according to literature (Takegawa et al., 2005; Canagaratna et al., 2007). Collection efficiency (CE), compensating for losses of the particles during their collection, is considered to be another extremely important parameter for quantification of the ACSM data. This quantity varies with



acidity, chemical composition, and water content of the particles (Matthew et al., 2008). Here we consider the effects of acidity and water content to be negligible based on the facts that the RH in the

sample line kept below 30% through a Nafion dryer and the aerosols were approximately neutralized in Guangzhou ($NH_4^+/NH_4^+{}_{predict} = 0.87$). Thus, only chemical composition was considered to affect CE in this work and we adopted a composition-dependent CE formulated by Middlebrook et al. (2012) (i.e., $CE = max (0.45, 0.0833 + 0.9167 \times ANMF$, where ANMF is the mass fraction of ammonium nitrate in NR-PM$_1$) instead of a widely used empirical value of 0.5. Furthermore, OA mass spectra were

deconvolved through Multilinear Engine (ME-2) and detailed deconvolution through ME-2 can be found in supplementary information (SI).

## 3 Results and discussion

### 3.1 Mass concentrations and chemical composition

Figure 1 shows the time series of meteorological conditions (relative humidity, temperature, pressure, wind speed and direction), NR-PM$_1$ and PM$_{2.5}$ mass concentrations, and NR-PM$_1$ composition ($SO_4^{2-}$, $NO_3^-$, $NH_4^+$, $Cl^-$, and organics) from 20 November 2017 to 5 January 2018. Overall, the 10-minute averaged mass concentration of NR-PM$_1$ ranged from 2.4 to 130 µg m$^{-3}$. The averaged concentration during the measurement period was $35.3 \pm 22.3$ µg m$^{-3}$, among which about half (17.5 µg

m$^{-3}$) was organics, followed by sulfate (7.0 µg m$^{-3}$), nitrate (6.0 µg m$^{-3}$), ammonium (4.6 µg m$^{-3}$), and chloride (0.5 µg m$^{-3}$). The NR-PM$_1$ concentration (from the ToF-ACSM) was correlated well with concentrations of both PM$_{2.5}$ (Pearson Correlation Coefficient ($R_p$) = 0.83, from BAM-1020) and PM$_{1.1}$ ($R_p$ = 0.86, from Anderson nine-stage sampler). Comparisons between ToF-ACSM and other instruments are detailed in Fig. S1. Overall, high NR-PM$_1$ mass concentration was observed after 22

December 2017 when the air was more stagnant according to the wind speed (Fig. 1).

     Moderate to severe pollution events were observed during the measurement period and we classified five pollution episodes (EP1~EP5) alongside with other periods defined as non-pollution based upon the NR-PM$_1$ mass concentrations in order to better understand the intrinsic mechanisms on the PM evolution. Each pollution episode was defined as a period that the NR-PM$_1$ mass concentration

continually increased from less than 20 µg m$^{-3}$ (10$^{th}$ percentile) to greater than 75 µg m$^{-3}$ (90$^{th}$ percentile) and then fell below 20 µg m$^{-3}$ again (Fig. 1d). For the entire study, organics and sulfate (Fig. 2) dominated NR-PM$_1$ mass concentration, consistent with previous studies conducted in the PRD region during autumn and winter (He et al., 2011; Huang et al., 2011; Qin et al., 2017). Fractions of 20% and



49% respectively for sulfate and organics were similar to those in Panyu (25% and 50% for sulfate and

organics respectively) and Shenzhen (28% and 46% for sulfate and organics respectively) but quite

differed from those in Kaiping (both 36%). This comparison reflects that the sulfate fraction increased

and the organic fraction simultaneously decreased from urban to rural areas (Table S1), which could be

likely attributed to different sources and chemical aging processes between urban and rural areas in the

PRD region. The obvious facts are that heavy industry sections, such as power plants and oil refineries

which emit a tremendous amount of sulfur dioxide, the primary precursor of sulfate, are mainly located

in non-urban areas. Similar urban-to-rural evolution of PM chemical composition was reported in

previous studies (Jimenez et al., 2009; Zhang et al., 2011; Li et al., 2017), providing further rationales

for performing field measurements under different source environments to investigate the chemical

evolution of PM.

Figure 3 shows a comparison of NR-PM$_1$ characteristics among our work in the PRD region and

previous wintertime studies conducted in several other Chinese megacities (Beijing, Nanjing,

Shijiazhuang, Lanzhou, Hong Kong) beyond PRD region (Sun et al., 2013; Xu et al., 2016; Sun et al.,

2016; Zhang et al., 2017; Huang et al., 2019). Compared to other locations, the averaged mass

concentration of NR-PM$_1$ in Guangzhou was much lower (35.5 μg m$^{-3}$), indicating that the air of PRD

region was relatively clean in terms of fine PM. The concentrations of NR-PM$_1$ in the abovementioned

cities were all composed of a large fraction of OA, with a substantial variation (37%~58%), indicating

the significant contribution of OA to fine particle mass loading over China. In addition, a remarkable

distinction on the fraction of secondary OA (SOA) in OA between southern and northern China was

found, that is, SOA dominated OA in southern China (A SOA to OA ratio of 0.66 in Nanjing, 0.59 in

Hong Kong, and 0.7 in Guangzhou), compared to a lower ratio of SOA to OA in northern China (0.31

in Beijing, 0.37 in Lanzhou, and 0.21 in Shijiazhuang). The increasing ratio of SOA to OA from the

North to the South is probably due to more favorable meteorological conditions such as solar radiation

and temperature for secondary chemical processes in southern China and more significant contribution

of coal combustion in northern China during wintertime (Sun et al., 2013; Sun et al., 2014; Sun et al.,

2018).

## 3.2 OA apportionment

Positive matrix factorization with ME-2 engine algorithm was employed to deconvolve OA into

four factors, including two primary OA components (HOA and COA) and two OOA components



(SVOOA and LVOOA) which are usually treated as SOA. The mass spectra and corresponding time series are depicted in Fig. 4.

### 3.2.1 Hydrocarbon-like OA (HOA)

A widely referred standard mass spectrum of HOA (Sun et al., 2013; Sun et al., 2016; Huang et al., 2019) derived by Ng et al. (2011) was introduced as an external constraint in this work, with an a-value of 0.3 being chosen to derive the final solution. The final mass spectrum of HOA was identified by the ion series representing $C_nH_{2n-1}^+$ ($m/z$ = 27, 41, 55, 69, 83, 97, typical tracers of cycloalkanes or unsaturated hydrocarbon) and $C_nH_{2n+1}^+$ ($m/z$ = 29, 43, 57, 71, 85, 99, typical tracers of alkanes). As shown in Fig. 4b, the concentration of HOA was well correlated with that of $NO_x$ during the measurement period when both concentrations were available, indicating considerable influences of traffic emissions on the HOA mass loading. Diurnal variations of HOA concentration for both pollution EPs and non-pollution period are depicted in Fig. 5. The variations for both mass concentrations of HOA and $NO_x$ were pronounced for pollution EPs (much higher concentrations at night), which can be attributed to human activities (e.g., traffic emission during rush hours). The averaged HOA concentration for pollution EPs ranged diurnally from 2.8 to 5.2 μg m$^{-3}$ with the maximum value (about 16% of total OA) found around midnight and the minimum value (12% of total OA) at noon (13:00). Meanwhile, the HOA concentration in pollution EPs increased rapidly from 3.1 μg m$^{-3}$ at ~17:00 to 5.2 μg m$^{-3}$ at around midnight and remained high afterward. Nocturnal rush hours corresponded to HOA concentration peak around 20:00 but could not account for the continuously high HOA concentration afterward. These consistent high concentrations were likably attributed to emissions of heavy-duty vehicles (HDV) which are only allowed to drive through the city after 22:00 until 6:00 the next day according to the traffic regulation enforced in Guangzhou. The pronounced impact of HDV on the concentrations of both HOA and $NO_x$ agrees with the previous study conducted in Panyu district, Guangzhou (Qin et al., 2017). Other possible reasons for high nocturnal HOA mass loading included lower boundary layer and frequent thermal inversion layer formed at night during winter. In comparison, the HOA concentration showed almost no variations during non-pollution period (even during rush hours) and remained extremely low (<1 μg m$^{-3}$ which was close to an estimated detection limit of 0.7 μg m$^{-3}$ for OA with the ToF-ACSM).

### 3.2.2 Cooking OA (COA)

Through factorization using PMF or ME-2 engine, COA was frequently deconvolved as a common OA component in urban areas (Sun et al., 2013; Lee et al., 2015; Sun et al., 2016; Qin et al., 2017). The


mass spectrum of COA deconvolved in this work was very similar to that of HOA except a higher $m/z$
55 to 57 ratio of 2.1 which is very close to the range of 2.2-2.8 reported from real cooking source
measurements (Mohr et al., 2012). The concentration of COA was correlated reasonably well ($R_p$= 0.86)
with m/z 55. As shown in Fig. 5, the diurnal profile of COA for non-pollution period showed a typical
bimodal pattern with a noon peak concentration of 1.6 μg m$^{-3}$ (16% of OA) at 13:00 during lunch time

and a night peak concentration of 4.3 μg m$^{-3}$ (33% of OA) at 19:00 during dinner time. A similar
bimodal pattern of COA diurnal profile for pollution EPs was found, with a much higher ratio of night
concentration peak (12.2 μg m$^{-3}$) to noon concentration peak (2.5 μg m$^{-3}$) than that for non-pollution
period (5 vs 2.7). In addition, compared to non-pollution period, the remarkably enhanced night
concentration peak of COA for pollution EPs was delayed from 19:00 to 21:00. Here three reasons are

attributed to the much higher peak concentration at night than at noon for COA: (1) more intensive
cooking activities at night than at noon in the Chinese cooking routine; (2) more adverse diffusion
conditions caused by a lower boundary layer and a frequent thermal inversion layer at night; (3) the
lower temperature at night which facilitates semi-volatile compounds from cooking emissions to
partition into particles. Furthermore, pollution EPs with remarkably enhanced and delayed night COA

concentration peaks corresponded to several important holidays such as Winter solstice festival (EP3),
Christmas (EP4), and New Year's Day (EP5), implying that festival-induced emissions have significant
impacts on local air pollution.

### 3.2.3 Oxygenated OA (OOA)

OOA, the generally accepted surrogate of SOA (Jimenez et al., 2009), characterized by a high peak

at $m/z$ 44 ($CO_2^+$) in the mass spectra, was almost exclusively contributed from electron ionization of
ketones, aldehydes, esters and carboxylic acids. Furthermore, OOA could be further deconvolved into
two subcomponents, SVOOA and LVOOA, based on different degrees of oxidation. SVOOA was
distinguished from LVOOA by a higher ratio of $f_{43}$ to $f_{44}$ (0.7 for SVOOA and 0.18 for LVOOA).

A high similarity between SVOOA and secondary inorganic aerosols (nitrate and sulfate) was

observed for the entire study (Fig. 4). It was found that Pearson correlation coefficient ($R_p$) between
SVOOA and nitrate (0.76) was higher than that between SVOOA and sulfate (0.61), consistent with the
trend reported by previous studies which attributed it to analogous high volatility and gas-particle
partitioning between SVOOA and nitrate (Aiken et al., 2009; DeCarlo et al., 2010; Zhang et al., 2011).
Interestingly, we found that the correlation between SVOOA and nitrate was much better for pollution

EPs than for non-pollution period ($R_p$ = 0.64 vs 0.34, Fig. 6d), which cannot be simply attributed to





similar volatility and photochemistry. Meanwhile, concentration of neither nitrate nor SVOOA was obviously dependent on temperature in this study. Furthermore, we found that the air during all pollution EPs was much more stagnant, supported by much lower wind speed than that during non-pollution period (Fig. 6f), implying that most nitrate and SVOOA were locally formed during

pollution EPs, while they likely originated from regional transport during non-pollution period. Since traffic emissions contribute largely to $NO_x$ and VOCs, a higher correlation between SVOOA and nitrate during pollution EPs than during non-pollution period likely stemmed from shared sources of precursors for the two aerosol components and implies that traffic emissions could significantly influence SVOOA and nitrate mass loading under stagnant meteorological conditions. These are further confirmed by the

fact that in general much higher correlations between SVOOA (or nitrate) and the traffic trace species (CO, HOA and $NO_x$) were found during pollution EPs than during non-pollution period (Table 1). As mentioned above, non-pollution period was always associated with high wind speeds which would induce strong air advection and make local emissions hardly accumulate, leading to a large fraction of $PM_1$ contributed from regional transport. These regionally transported nitrate and SVOOA were usually

less correlated with traffic tracers because of more complicated sources for SVOOA and nitrate outside urban Guangzhou, and different influences of secondary processes on these species through transport. A previous laboratory study suggested that photooxidation of traffic related emissions leads to substantial amounts of SOA (Weitkamp et al., 2007) which in our case are likely locally formed SVOOA. The diurnal profiles of SVOOA for non-pollution period and pollution EPs showed explicit distinctions.

During pollution EPs, the concentration of SVOOA showed a pronounced diurnal variation which in general formed a flat trough around noon and reached the maximum value around midnight. In comparison, diurnal profile of SVOOA concentration for non-pollution period, however, was much flatter, with two weak peaks likely reflecting photochemical oxidation and local anthropogenic emissions respectively. These two different diurnal patterns of SVOOA concentration suggest

accumulation of local anthropogenic emissions under stagnant air condition during pollution EPs could largely influence mass loading of SVOOA.

In contrast to SVOOA, the concentration of LVOOA showed a better correlation with that of sulfate ($R_p = 0.85$) than with nitrate ($R_p = 0.77$), which is likely attributed to similarly low volatility of both LVOOA and sulfate. Diurnal profile of LVOOA concentration showed roughly no variation with

340 slightly elevated values during afternoon for both pollution EPs and non-pollution period, demonstrating that LVOOA was not significantly influenced by local emissions but by aging processes



in particular photochemistry. Prolonged aging of SOA leads to the high oxidation degree of LVOOA which should be considered as a kind of regional pollutant or background aerosol (Li et al., 2013; Li et al., 2015; Sun et al., 2016; Qin et al., 2017), while freshly formed SOA from local emissions remains a

lower oxidation state which more readily becomes the SVOOA component as is discussed in this section.

### 3. 3 Diurnal profiles

In the previous section, we have discussed diurnal variations of the four deconvolved OA components and here important variations of other species and meteorological conditions are detailed to

better understand evolution of NR-PM$_1$ species and pollution characteristics in Guangzhou (Fig. 5). Similar to SVOOA, remarkably different diurnal profiles of nitrate concentration between pollution EPs and non-pollution period were observed. For non-pollution period, diurnal nitrate concentration varied little with slight maximum occurring at 13:00, implying the influence of photochemistry on nitrate formation. For pollution EPs, however, nitrate concentration showed a much more pronounced diurnal

variation, that is, nitrate mass loading kept low around noon to afternoon and then increased steadily during the night until it reached the maximum at 9:00 in the next day. Note that nitrate concentration increased from dusk to early morning with similarly increasing RH and oppositely decreasing temperature. Meanwhile, the nocturnal NO$_x$ concentration was 67% more than that during daytime (Table 2). Thus, higher RH, more abundant NO$_x$, and lower temperature during nighttime facilitate

aqueous reactions and gas-to-particle partitioning between gas phase nitric acid and ammonium nitrate, which played important roles in nocturnal nitrate formation during pollution EPs, consistent with the previous study (Xue et al., 2014). In addition, the morning nitrate peak at 9:00 can be attributed to a synergy of high NO$_x$ emissions during rush hours and most favorable conditions for ammonium nitrate formation during 8:00~9:00 am (i.e., low temperature, high RH). Our results strongly demonstrate

importance of local anthropogenic emissions and aqueous reactions in nitrate accumulation under stagnant air condition during pollution EPs.

The diurnal concentration of sulfate showed very slight variation for both pollution EPs and non-pollution period, with the averaged daytime concentration being almost the same as that at night. This indicates that the sulfate concentration was always less influenced by local anthropogenic

emissions due to the fact that power plants, the most important source of sulfur dioxide (SO$_2$), are rarely located in urban Guangzhou (Bian et al., 2019). Thus, the sulfate concentration in this study should be



largely determined by regional transport during both pollution EPs and non-pollution period, which can also be supported by similar diurnal profiles of $SO_2$ (Fig. S2). Interestingly, the diurnal variations of $O_3$ and $RO_2^*$ ($\Sigma RO_2{}^{\bullet} + HO_2$) for pollution EPs and non-pollution period showed a distinct

daytime-to-nighttime pattern. The daytime concentrations of the two species during pollution EPs were higher than those during non-pollution period while the nighttime concentrations during pollution EPs were lower than those during non-pollution period. This variation pattern differed from all other NR-PM$_1$ species which always showed higher concentrations during pollution EPs than those during the non-pollution period within a day. The lower nighttime concentrations of $O_3$ and $RO_2^*$ during pollution

EPs were probably attributed to the enhanced consumption of $O_3$ and $RO_2^*$ due to elevated $NO_x$ and VOCs concentrations at night. Our results suggest that $O_3$ and $RO_2^*$ were important nocturnal oxidants during pollution events.

### 3.4 Chemical evolution

Figure 7 depicts the dependence of mass concentrations and fractions of NR-PM$_1$ components on

NR-PM$_1$ mass loading. The mass concentrations of all the NR-PM$_1$ components increased almost linearly with increase of the NR-PM$_1$ mass concentration. However, various trends were detected for the fractions of different NR-PM$_1$ components. For example, organics were the dominant component of NR-PM$_1$ with an increasing fraction from 44% to 57% as the NR-PM$_1$ mass concentration increased up to > 90 $\mu g\ m^{-3}$ (Fig. 7b). The fractions of HOA and COA, considered as primary OA, varied in a similar

way, that is, both fractions increased up to 11% for HOA and 15% for COA as NR-PM$_1$ mass concentration increased from ~35 $\mu g\ m^{-3}$ to > 90 $\mu g\ m^{-3}$. The variations of the fractions of SOA species and SIA species with NR-PM$_1$ mass loading were much more complicated. To have an overall insight on chemical evolution of the aerosols, we compare dependences of the ratio of secondary particulate matters (SPM = SVOOA + LVOOA + sulfate + nitrate + ammonium) to primary particulate matters

(PPM = HOA + COA + chloride) and the concentration of $RO_2^*$ and $O_3$ on NR-PM$_1$ mass concentration (Fig. 8). For the whole period, all the three quantities followed consistent and clear trends that they all initially increased with increase of NR-PM$_1$ mass concentration until all reached peak values at a NR-PM$_1$ concentration of ~35 $\mu g\ m^{-3}$, and subsequently decreased with increase of the NR-PM$_1$ mass loading. Interestingly, the trends for the three quantities during non-pollution period were consistent

with the initial monotonic increase when NR-PM$_1$ mass concentration was below 35 $\mu g\ m^{-3}$ (Fig. 8c), a concentration value that was rarely exceeded during this period, while only if pollution EPs were



considered, the trends became monotonic decrease with NR-PM$_1$ mass concentration up to ~95 μg m$^{-3}$ (Fig. 8b). During non-pollution period when strong advection induced by high wind speed facilitated dilution and diffusion of local primary pollutants, the SPM/PPM ratio increased with increasing

concentrations of photochemical products (i.e., O$_3$ and RO$_2$*) and NR-PM$_1$, strongly suggesting that secondary processes especially photochemistry were the main drivers for NR-PM$_1$ accumulation during this period. Under this circumstance, interestingly, the overall increase of SPM fraction (or SPM/PPM) with increase of NR-PM$_1$ mass concentration was consistent with increasing mass fractions of LVOOA and nitrate, yet opposite from decrease of SVOOA fraction (Fig. 7b and Fig. S3). The decrease of

SVOOA fraction with increase of NR-PM$_1$ mass concentration corresponded to increase of LVOOA fraction, implying progressive conversion of SVOOA to LVOOA during the aerosol aging processes. During pollution EPs when stagnant air condition supported by low wind speeds facilitated accumulation of local primary pollutants, the SPM/PPM ratio together with concentration of O$_3$ and RO$_2$*, was observed to drop with increasing mass concentration of NR-PM$_1$, indicating that production

of NR-PM$_1$ was more driven by primary emissions rather than secondary processes under this circumstance. Thus, our results indicate totally different intrinsic mechanisms responsible for NR-PM$_1$ accumulation between pollution EPs and non-pollution period.

**3.5 Oxidation degree and peroxy radical tracer**

**3.5.1 Oxidation degree**

Oxidation degree represents the extent to which aerosols (OA and SOA) are oxidized. Figure 9 depicts locations of both OA and SOA from this study in $f_{44}/f_{43}$ space. The triangle area (enclosed by two black dash lines and the $f_{43}$ axis) for ambient OOA (SOA) was defined by Ng et al. (2010) and results from several laboratory studies with or without aerosol seeds (Bahreini et al., 2005; Liggio et al., 2005; Liggio et al., 2006; Weitkamp et al., 2007) are also shown for comparison. The $f_{44}$ and $f_{43}$ for each

SOA point were calculated through an algorithm reported in Canonaco et al. (2015). Most of OA samples in this study were located inside the triangle area, indicating that OA is mainly composed of SOA which was consistent with the finding from ME-2 analyses, that is, SOA contributes about 70% to total OA. Interestingly, SOA points in $f_{44}/f_{43}$ space showed a strong linear trend that $f_{44}$ increased with decreasing $f_{43}$, leading to a large span of $f_{44}$ and $f_{43}$ in SOA which then requires more than one SOA

factor to explain such a large variation. This result quite differs from some other studies in northern China where usually only one SOA factor was deconvolved (Sun et al., 2012; Sun et al., 2013; Huang et





al., 2019). The substantial differences on SOA factorization between Guangzhou and other cities in northern China can be likely attributed to much higher oxidative atmosphere in Guangzhou. In fact, a previous study has shown that the second highest OH concentration was observed in the PRD region around the world (Rohrer et al., 2014). In addition, the majority of SOA points were well distributed around the connection line of SVOOA and LVOOA (unimodal residual), indicating that SOA points were well captured by ME-2 engine in our study (Canonaco et al., 2015). Thus, the deconvolution of SVOOA and LVOOA in this study well represented the observed large variation of $f_{44}$ and $f_{43}$ for SOA in Guangzhou.

Most SOA points were located closely to the right side of the triangle, which was quite different from the OA points. Interestingly, the shape of SOA points in $f_{44}/f_{43}$ space formed roughly a tilted triangle which means both average values and variation ranges of $f_{43}$ remarkably decreased with increasing $f_{44}$. The triangle suggests that aging processes result in substantially similar OOA components regardless of their original precursors (Ng et al., 2010). The $f_{44}$ values of SOA in this study were much higher than those of SOA generated in laboratory, which should be attributed to limited residence time in chamber or much higher mass loadings of laboratory-produced aerosols which facilitate gas-to-particle partitioning of semi-volatile organic compounds (SVOCs) and heterogeneous reactions (Liggio et al., 2006; Shilling et al., 2009; Ng et al., 2010; Ziemann et al., 2012). Figure 9 also shows that the $f_{44}$ and $f_{43}$ values of SOA generated from laboratory m-xylene oxidation and from aged diesel exhaust were closest to those of the SOA and modeled SVOOA from this study, implying that traffic emissions (in particular from diesel combustion) and aromatic compounds are likably precursors for SOA in Guangzhou. This result also agrees with numerous studies conducted in urban areas (Lee et al., 2015; Sun et al., 2016; Qin et al., 2017).

### 3.5.2 Peroxy radical tracer

Field and laboratory studies have shown that photochemical oxidation contributes significantly to SOA formation. The odd oxygen ($O_x = O_3 + NO_2$) was proven to be a robust indicator of photochemical intensity and was therefore adopted in numerous studies to discuss SOA formation (Herndon et al., 2008; Zhou et al., 2014). However, these studies employing $O_x$ only focused on daytime SOA formation while the nocturnal SOA formation was generally ignored due to the limitation of $O_x$. Meanwhile, interference form directly emitted $NO_2$ will also lead to some uncertainties. Here we introduce peroxy radicals ($RO_2^* = \Sigma RO_2^\bullet + HO_2$) as a reference tracer to discuss both daytime and nocturnal SOA



formation.

Oxidation of atmospheric VOCs is initiated by important oxidants (e.g., OH, $O_3$ or $NO_3$) to form

alkyl radicals ($R^\bullet$) which are subsequently oxidized to form peroxy organic radicals ($RO_2^\bullet$) and alkoxy

radicals ($RO^\bullet$) (Kroll et al., 2008; Ziemann et al., 2012). The simplest $RO_2^*$, $HO_2$, is formed in the

atmosphere via three pathways: (1) from reactions of OH radicals with ozone or CO; (2) from oxidation

of VOCs; (3) from photolysis of formaldehyde (Levy et al., 1971; Ziemann et al., 2012; Sheehy et al.,

2010; Stone et al., 2012; Griffith et al., 2013; Wang et al., 2014). During daytime, $RO_2^*$ radicals are

mainly photochemical products since their formation depends highly on solar radiation or OH radical

(Sheehy et al., 2010; Stone et al., 2012; Griffith et al., 2013; Wang et al., 2014). In this work,

concentrations of $RO_2^*$ and $O_x$ were found to be highly correlated during daytime, providing an

additional rationale for choosing $RO_2^*$ as a tracer for SOA due to photochemical oxidation (Fig. S4).

During nighttime, however, nighttime oxidation of VOCs initiated by ozone or nitrate radicals

dominates formation of $RO_2^*$ radicals especially in urban area, which has been proven in Volkamer et

al. (2010) and Stone et al. (2014). It is hence that $RO_2^*$ can served as a tracer for photochemical

induced SOA formation during daytime and SOA formation induced by nocturnal gas phase oxidation

of VOCs during nighttime.

Figure 10 shows the variations of SOA concentration with $RO_2^*$ concentration under different

scenarios (non-pollution daytime period, pollution daytime EPs, non-pollution nighttime period, and

pollution nighttime EPs). The SOA concentration increased with the $RO_2^*$ concentration for both

non-pollution daytime and nighttime periods with both moderate correlation coefficients ($R^2= 0.41$),

indicating that formation and growth of SOA could be attributed to photochemical oxidation during

non-pollution daytime while nocturnal gas phase oxidation of VOCs led to SOA accumulation during

non-pollution nighttime. Thus, it can be concluded that gas phase oxidation was responsible for SOA

formation during non-pollution periods. Meanwhile, SOA oxidation degree, represented by $f_{44}$ in SOA,

increased in general with increasing SOA concentration during non-pollution periods, implying that a

higher oxidative condition simultaneously led to generation of SOA and conversion of SVOOA to

LVOOA (Fig. 7b). In contrast to the well correlations between SOA and $RO_2^*$ concentrations during

non-pollution period, such correlations were not seen during the pollution EPs for both daytime and

nighttime scenarios (Fig. 10); instead, an opposite trend that $f_{44}$ in SOA decreased with increase of SOA

concentration was observed (Figs. 10c-d). These results imply that other mechanisms besides gas phase



oxidation were responsible for the formation of SOA that is less oxidized during pollution EPs. Rapid accumulation of local emissions under stagnant conditions not only produced a substantial amount of VOCs that compete for oxidants but also provided high particle volumes for heterogeneous reactions,

probably leading to poor correlations between SOA concentration and $RO_2^*$ concentration, and also less-oxidized SOA during pollution EPs. In addition, SOA formed via heterogeneous reactions is likely more related to PM concentration and less influenced by $RO_2^*$ or gas phase oxidation.

To further explore mechanisms that can explain SOA formation during pollution EPs, a plot of SOA concentration as a function of $RO_2^*$ concentration for segmental $NR\text{-}PM_1$ mass concentrations is

500 shown in Fig. 11. A total of six segments of concentrations were set, with concentrations smaller than 30 and larger than 70 up to about 110 µg m$^{-3}$ being the lowest and highest segments respectively, and with an interval of 10 µg m$^{-3}$ between 30 and 70 µg m$^{-3}$ (Fig. 11a). For comparison, dependence of SOA concentration on $RO_2^*$ concentration during non-pollution period is also included in Fig. 11. As discussed above, the overall correlation between SOA and $RO_2^*$ concentrations was poor without a

505 clear trend between the two quantities (Figs. 10c-d). However, they were reasonably correlated and showed reasonable trend that SOA concentration increased with increasing $RO_2^*$ concentration in all segments except for the lowest one, suggesting that gas phase oxidation still played important roles in SOA formation. Poor correlation between SOA and $RO_2^*$ concentrations in the lowest segment might be due in part to substantial scattering of the data. Under a constant $RO_2^*$ concentration, interestingly,

more SOA was formed with increasing $NR\text{-}PM_1$ concentration. Here, for simplicity, we define that the slope and intercept of linear regression between SOA and $RO_2^*$ concentrations in each segment when $NR\text{-}PM_1$ ranged from 30-40 to > 70 µg m$^{-3}$ represented respectively SOA formed due to $RO_2^*$ and SOA contributed from other pathways. We assume that $RO_2^*$ can roughly represent the amount of gas phase oxidation products of VOCs which seems to be reasonable. Values of the slopes were observed to

increase with increasing $NR\text{-}PM_1$ mass loading, likely suggesting more gas phase oxidation products were partitioned into particles under high PM mass loading (Odum et al., 1996). Furthermore, substantial intercept values were obtained from the linear regressions during pollution EPs, while they approached almost zero during non-pollution period. As mentioned above, the intercepts represent SOA contributed from other pathways. The exact pathways were currently unknown and could not be

identified from this study due to obviously insufficient measurement data. However, they are most likely attributed to heterogeneous or/and multiphase reactions on particle surfaces or inside particles. The intercepts increased from 7.05 to 17.5 µg m$^{-3}$ when $NR\text{-}PM_1$ mass loading increased from 30-40





to >70 μg m$^{-3}$, indicating enhanced heterogeneous/multiphase reactions with increase of PM mass loading.

## 4 Conclusions

A field campaign employing ToF-ACSM for measurements of NR-PM$_1$ chemical composition was conducted from 20 November 2017 to 5 January 2018 at an urban site in Guangzhou, China. The reliability of the ToF-ACSM was confirmed by the well correlation between mass concentrations of

NR-PM$_1$ measured by this instrument and those of PM$_{2.5}$ and PM$_{1.1}$ measured from other filter-based methods. Chemical composition of NR-PM$_1$ at this site was dominant by organics, followed by sulfate, nitrate, and ammonium, and only an insignificant fraction (1-2%) of chlorine was measured. We classified five pollution episodes (EPs) according to mass concentration of NR-PM$_1$. Mass fraction of organics was higher during pollution EPs than during non-pollution period, corresponding to a decrease

and an increase of sulfate fraction respectively for the two periods. Our results together with other previous studies show increasing SOA/OA mass concentration ratio across China from the North to the South.

Positive Matrix Factorization (PMF) with multilinear engine (ME-2) algorithm was employed to deconvolve OA into four factors including hydrocarbon-like OA (HOA, 12%), cooking OA (COA,

18%), semi-volatile oxygenated OA (SVOOA, 30%), and low-volatility oxygenated OA (LVOOA, 40%) according to the mass spectra acquired by the ToF-ACSM during the study. One primary OA, HOA, was found to originate from heavy-duty vehicles (HDV) emissions after midnight during which those vehicles are allowed to enter urban Guangzhou according to the traffic regulation. Another primary OA, COA, was found to be significantly contributed by nocturnal cooking activities during pollution EPs.

Those activities were extended during festival celebrations, leading to an obviously delayed peak concentration for COA. Both cooking and traffic emissions contributed significantly to nocturnal PM accumulation because they emitted not only primary aerosols (COA and HOA) but also substantial amounts of precursors for SOA or nitrate. Concentrations of SVOOA and nitrate were correlated well with traffic tracers (i.e., CO, HOA and NO$_x$) during the pollution EPs, suggesting that SVOOA and

nitrate were significantly contributed from local traffic emissions under stagnant meteorological conditions and that they originated from shared precursor sources.

Concentrations of HOA, COA, nitrate, and SVOOA showed much more pronounced diurnal variations during pollution EPs since all four species were largely influenced by local anthropogenic


emission under stagnant conditions. For comparison, both sulfate and LVOOA showed little diurnal

variations during both non-pollution period and pollution EPs as they were contributed form regional

pollutants and were barely influenced by local anthropogenic emissions. In addition, nocturnal

concentrations of both $O_3$ and $RO_2^*$ during pollution EPs were lower than those during non-pollution

period, implying that both $O_3$ and $RO_2^*$ could serve as important nocturnal oxidants during pollution

EPs. Chemical evolution of NR-PM$_1$ suggests that different intrinsic mechanisms were responsible for

NR-PM$_1$ accumulation between pollution EPs and non-pollution period. During non-pollution period,

the SPM/PPM ratio increased with increasing concentrations of photochemical products (i.e., $O_3$ and

$RO_2^*$) and NR-PM$_1$ concentration, strongly suggesting that secondary processes especially

photochemistry were the main mechanism for NR-PM$_1$ accumulation. During pollution EPs, the

SPM/PPM ratio, together with concentration of $O_3$ and $RO_2^*$, was observed to drop with increasing

NR-PM$_1$ mass concentration, indicating that production of NR-PM$_1$ was more driven by primary

emissions rather than secondary processes. The $f_{44}/f_{43}$ space, proposed by Ng. et al. (2010), was

employed to investigate oxidation degree of OA and SOA and the results suggested that two OOA

factors were needed to cover a wide range of $f_{44}$ and $f_{43}$ in SOA in Guangzhou. Furthermore, we

conclude that traffic-emitted VOCs and aromatic compounds were most likably SOA precursors in

Guangzhou by comparing SOA from our measurements with those from other laboratory studies.

      Peroxy radicals $RO_2^*$ were measured in this study and were used as a tracer for gas phase

oxidation to explore SOA formation during both daytime and nighttime, advantaging over $O_x$ which can

only be used as an indicator of daytime photochemistry in addition to significant interference by directly

emitted $NO_2$ especially in urban area. During non-pollution periods, SOA concentration increased with

$RO_2^*$ concentration for both daytime and nighttime and we conclude that formation and growth of SOA

could be attributed to photochemical oxidation during daytime while nocturnal gas phase oxidation of

VOCs led to SOA accumulation during nighttime. We also found that SOA oxidation degree increased

in general with increasing SOA concentrations, implying that a higher oxidation condition

simultaneously led to generation of SOA and conversion of SVOOA to LVOOA. During pollution EPs,

however, an overall opposite trend was observed, that is, $f_{44}$ in SOA decreased with increase of SOA

concentration. In addition, the overall correlations between SOA and $RO_2^*$ concentrations were poor for

both daytime and nighttime. The reasons for the above poor correlations are attributed to other

mechanisms besides gas phase oxidation which are responsible for SOA formation during pollution EPs.

Possible mechanisms were explored by dividing PM$_1$ mass loadings into six segments and linear

regression in each segment was made between SOA and $RO_2^*$ concentrations. The results showed that individual correlations were reasonably good except for the lowest segment due to data scattering, indicating that gas phase oxidation still play important roles. Values of slopes from linear regressions of those reasonable correlations were then found to increase with increasing $NR\text{-}PM_1$ mass loading, suggesting that more gas phase oxidation products of VOCs were partitioned into particles under high

PM mass loading. Furthermore, substantial intercept values were found from the above linear regressions in contrast to almost zero intercept for non-pollution period, strongly suggesting that other mechanisms besides gas phase oxidation contributed significantly to SOA formation. We speculate that those other mechanisms are most likably heterogeneous or/and multiphase reactions.

**Acknowledgement**

This work was funded by the National Key Research and Development Program of China (2016YFC0202205, 2017YFC0210104), the National Natural Science Foundation of China (41875152, 91644225, 21577177), the National Science Fund for Distinguished Young Scholars (41425020), and the National Natural Science Foundation as a key project (41530641, 41630422). J.Z. acknowledges

funding support from the "111 plan" Project of China (Grant B17049), Scientific and Technological Innovation Team Project of Guangzhou Joint Research Center of Atmospheric Sciences, China Meteorological Administration (Grant No. 201704). The authors thank Prof. Xinhui Bi and Jingjing Feng for their support in the field study.

**Author contributions.** SZ and XW designed this study. SZ, JG, MC, WZ, CY, XX and WS conducted the experiments. JG, SZ and JZ wrote the paper. YS, WH, YH, ZZ, PC, QF, JH, SF, XW were involved in the data analysis and scientific discussions to the paper.

**Data availability.** All the data presented are available from the corresponding author upon reasonable
request.

**Competing interests.** The authors declare no competing financial interests.





**Table 1.** Pearson correlation for traffic tracers (CO, HOA and NO$_x$) with SVOOA (nitrate and sulfate) during different periods.

| Species | Period | CO | HOA | NO$_x$ |
|---|---|---|---|---|
| SVOOA | Pollution EPs | 0.64 | 0.70 | 0.81 |
| | Non-pollution | 0.37 | 0.55[a] | 0.63 |
| | Entire study | 0.72 | 0.82 | 0.82 |
| NO$_3^-$ | Pollution EPs | 0.53 | 0.61 | 0.52 |
| | Non-pollution | 0.09 | 0.45[a] | 0.22 |
| | Entire study | 0.57 | 0.75 | 0.59 |
| SO$_4^{2-}$ | Pollution Eps | 0.35 | 0.39 | 0.00 |
| | Non-pollution | 0.28 | 0.45[a] | 0.12 |
| | Entire study | 0.47 | 0.59 | 0.28 |

[a] Pearson correlations involving HOA for non-pollution period may have nonnegligible uncertainty since HOA concentration in non-pollution period is near method detection limit (MDL) of ToF-ACSM for OA.



**Table 2.** Overview of meteorological conditions, trace gases, peroxy radical, and NR-PM$_1$ components during day and night.

|  | Non-pollution | | Pollution | | Entire study | |
|---|---|---|---|---|---|---|
|  | Day | Night | Day | Night | Day | Night |
| ***Gas&radical(ppb)*** | | | | | | |
| O$_3$ | 20.3 | 10.4 | 27.5 | 7.2 | 22.6 | 9.7 |
| NO$_x$ | 28.6 | 34.2 | 43.1 | 72.7 | 31.4 | 41.5 |
| RO$_2$* | 0.15 | 0.12 | 0.17 | 0.11 | 0.16 | 0.11 |
| ***Meteorological condition*** | | | | | | |
| T(°C) | 16.6 | 15.4 | 19.1 | 17.6 | 17.5 | 16.2 |
| RH (%) | 48.3 | 52.9 | 41.0 | 45.6 | 45.6 | 50.1 |
| WS(m s$^{-1}$) | 3.1 | 3.2 | 1.6 | 1.4 | 2.5 | 2.5 |
| ***NR-PM1(µg m$^{-3}$)*** | | | | | | |
| Org | 9.0 | 10.0 | 23.8 | 32.0 | 15.0 | 19.0 |
| SO$_4^{2-}$ | 4.6 | 4.9 | 9.9 | 10.2 | 6.8 | 7.1 |
| NO$_3^-$ | 3.5 | 3.2 | 9.6 | 9.7 | 6.0 | 5.9 |
| NH$_4^+$ | 3.1 | 3.1 | 6.5 | 6.6 | 4.5 | 4.5 |
| Cl$^-$ | 0.2 | 0.3 | 0.6 | 0.9 | 0.4 | 0.5 |
| HOA | 0.7 | 0.8 | 3.4 | 4.7 | 1.8 | 2.4 |
| COA | 1.3 | 2.0 | 2.0 | 7.2 | 1.6 | 4.1 |
| SVOOA | 2.5 | 3.0 | 7.3 | 9.6 | 4.4 | 5.7 |
| LVOOA | 4.5 | 4.3 | 10.7 | 10.0 | 7.1 | 6.6 |





**Figure captions**

**Figure 1.** Temporal variation of (a) relative humidity, temperature and pressure; (b) wind speed and direction (color-contoured); (c) total NR-PM$_1$ & PM$_{2.5}$; and (d) speciated concentrations of NR-PM$_1$. EP1~EP5 are classified as pollution episodes (EP1: 6~8 December, EP2: 9~13 December, EP3: 21~23 December, EP4: 25~30 December, EP5: 31 December ~ 5 January), see text for details.

**Figure 2.** The average fraction of each chemical composition of (a) NR-PM$_1$ (OA, nitrate, sulfate, ammonium, and chloride) and (b) organic aerosols (LV-OOA, SV-OOA, HOA, and COA).

**Figure 3.** Comparison of submicron aerosols among several megacities in China during winter including Guangzhou (this study) and five other cities (Beijing: Sun et al., 2013; Nanjing: Zhang et al., 2017; Shijiazhuang: Huang et al., 2019; Lanzhou: Xu et al., 2016; Hong Kong: Sun et al., 2016). Red bars represent the fraction of OA to NR-PM$_1$ and blue bars represent the fraction of SOA to OA. Green diamonds represent average concentration of NR-PM$_1$ in each city.

**Figure 4.** The mass spectra and time series of the four OA components (HOA, COA, SVOOA, and LVOOA).

**Figure 5.** Diurnal profiles of NR-PM$_1$ species, trace gases, radicals, and meteorological conditions. Dash lines and solid lines represent the averaged values during non-pollution period and pollution EPs.

**Figure 6.** Correlations between OOA and SIA, along with correlation between nitrate and sulfate, and the wind speed during pollution EPs and non-pollution period. (a) LVOOA vs. NO$_3^-$; (b) LVOOA vs. SO$_4^{2-}$; (c) NO$_3^-$ vs. SO$_4^{2-}$; (d) SVOOA vs. NO$_3^-$; (e) SVOOA vs. SO$_4^{2-}$. Blue circles represent pollution EPs while gray crosses represent non-pollution period. (f) Box plot of wind speed in non-pollution period and pollution EPs. Whiskers are 10th and 90th percentile; the top, median and bottom lines of box represent 75th, 50th and 25th percentile respectively. Red dots are the averaged wind speed for each scenario.

**Figure 7.** Dependences of (a) mass concentration and (b) mass fraction of NR-PM$_1$ components on



NR-PM$_1$ mass loading. The data are plotted in an interval of 10 μg m$^{-3}$ of NR-PM$_1$. The error bars are standard deviations of each NR-PM$_1$ species.

**Figure 8.** Dependences of SPM/PPM ratio, concentrations of the atmospheric oxidants (O$_3$ and RO$_2$*) on NR-PM$_1$ mass loading for (a) entire study, (b) pollution EPs and (c) non-pollution period. The data are plotted in an interval of 10 μg m$^{-3}$ NR-PM$_1$.

**Figure 9.** Plot of $f_{44}$ vs $f_{43}$ for this study, along with several laboratory studies ([a] Bahreini et al., 2005; [b] Liggio et al., 2005; [c] Weitkamp et al., 2007). SOA are color-coded by OA mass concentration and the size of a circle is proportional to corresponding NR-PM$_1$ mass concentration. The triangle area (enclosed by two black dash lines and the $f_{43}$ axis) for ambient OOA (SOA) was defined by Ng et al. (2010).

**Figure 10.** SOA concentration as a function of RO$_2$* concentration during different scenarios (non-pollution daytime period, pollution daytime EPs, non-pollution nighttime period, and pollution nighttime EPs). SOA concentrations are color-coded by $f_{44}$ in SOA.

**Figure 11.** (a) Scatter plots of SOA and RO$_2$* during pollution EPs at different NR-PM$_1$ mass concentration segments, and (b) Correlation coefficients, slopes, and intercepts of linear regressions between SOA and RO$_2$* for NR-PM$_1$ mass concentration segments ranging from 30-40 to > 70 μg m$^{-3}$.





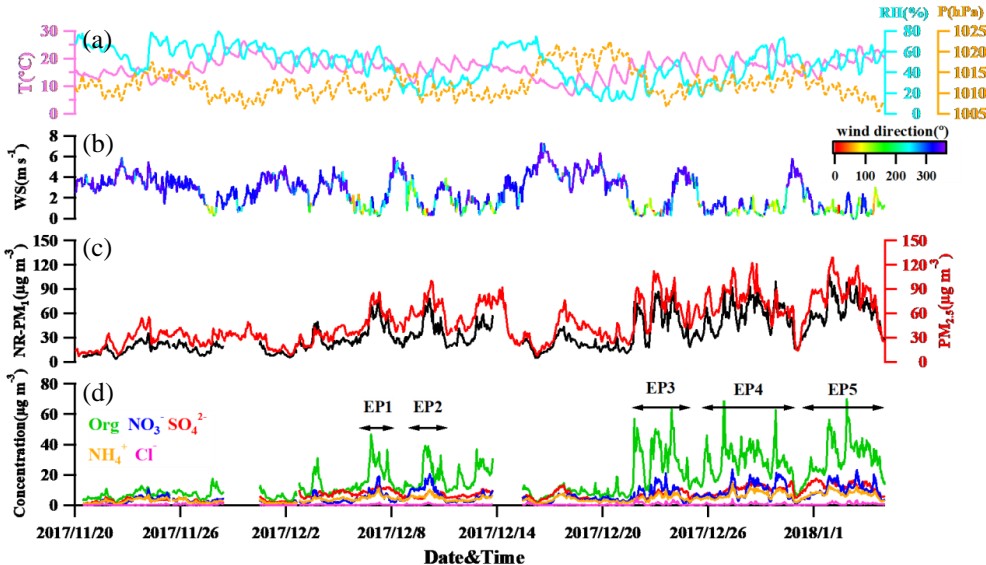

**Fig. 1.**





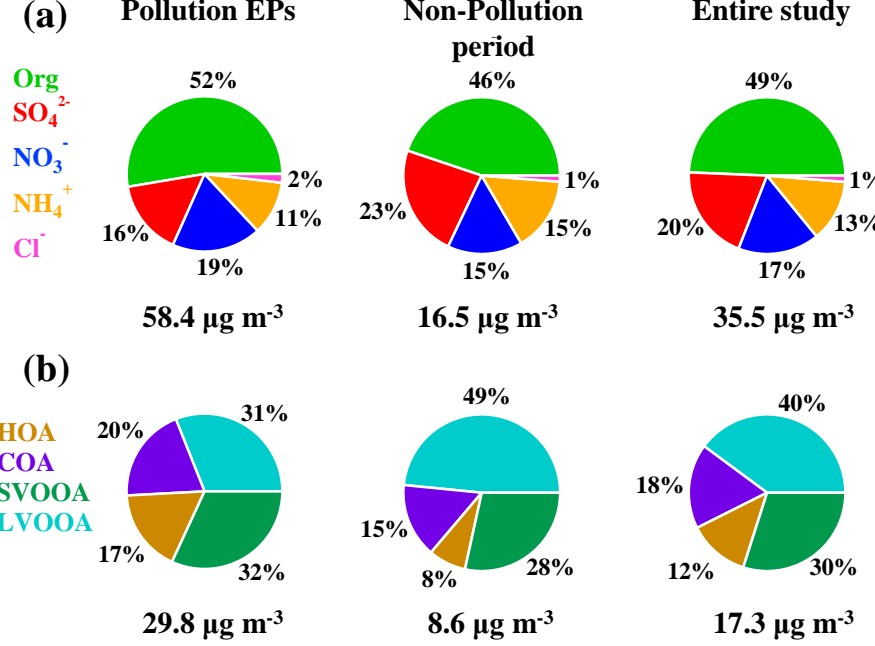

**Fig. 2.**





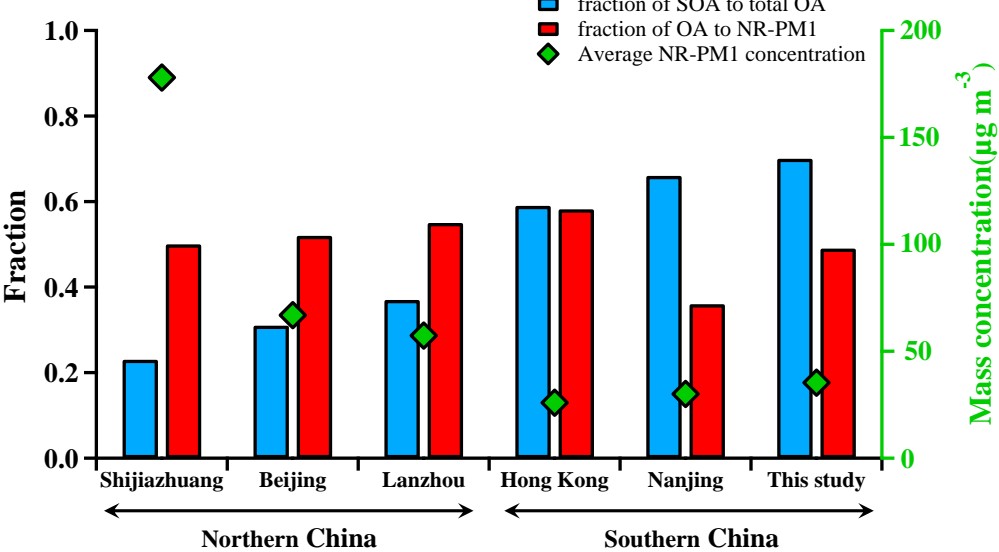

Fig. 3.



**Fig. 4.**



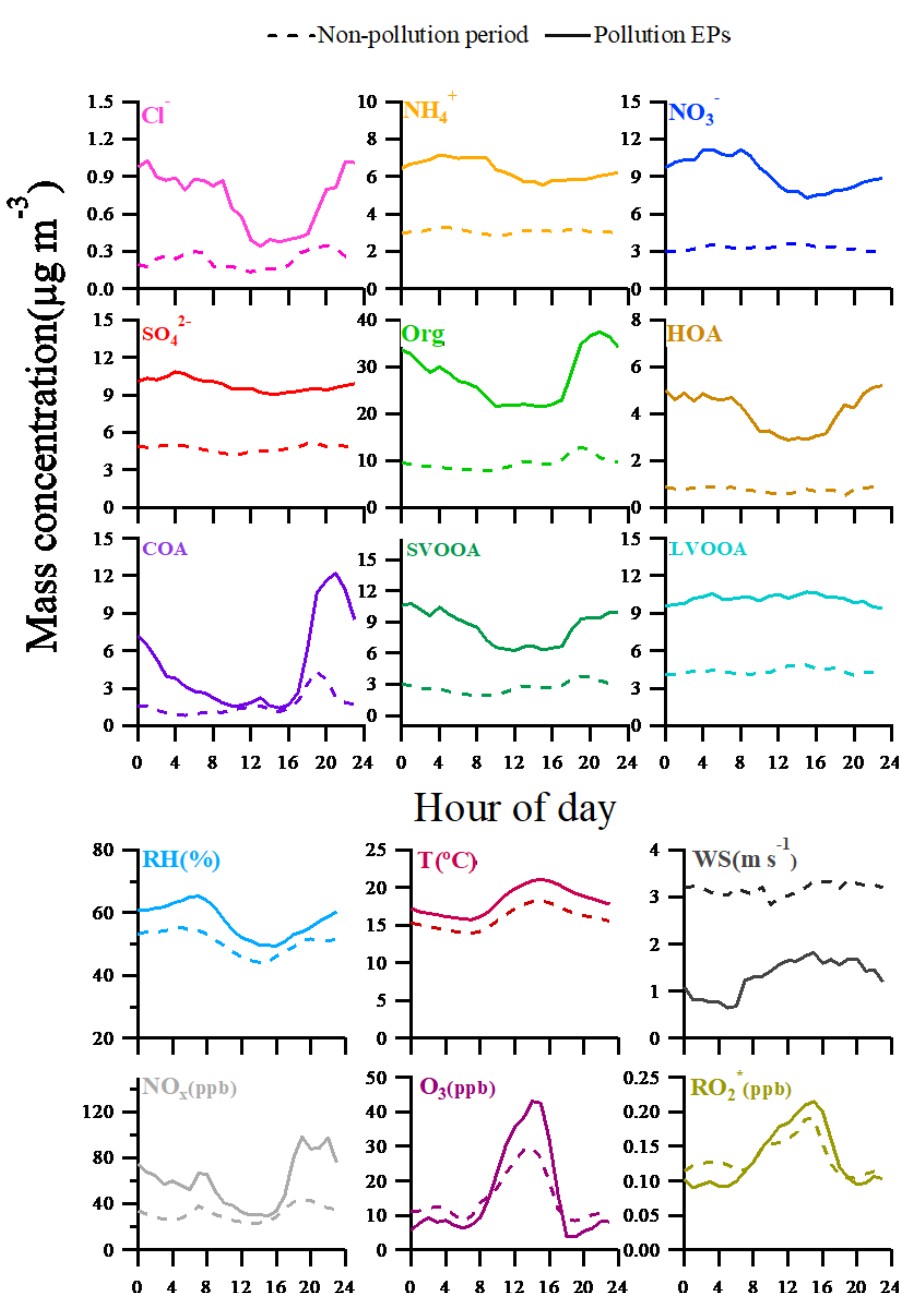

**Fig. 5.**





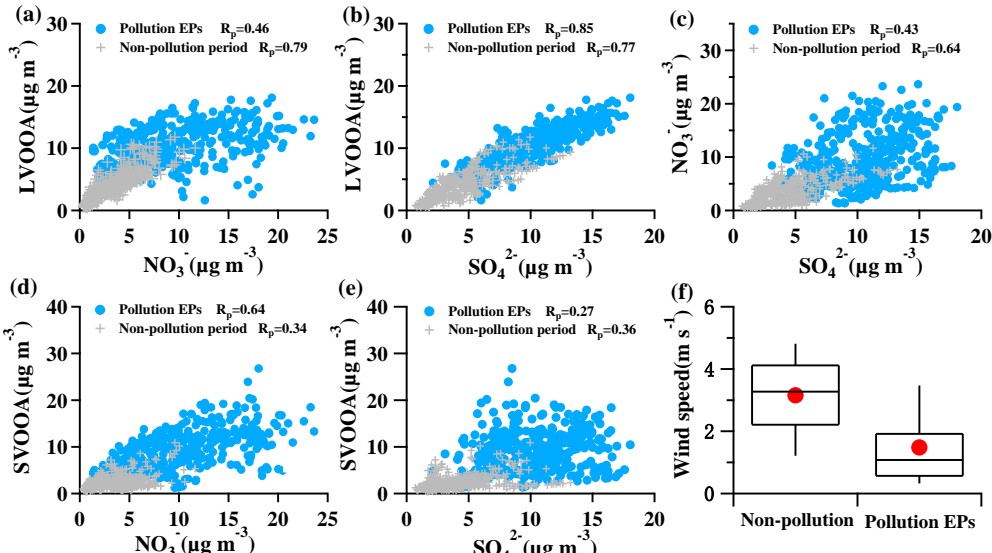

Fig. 6.





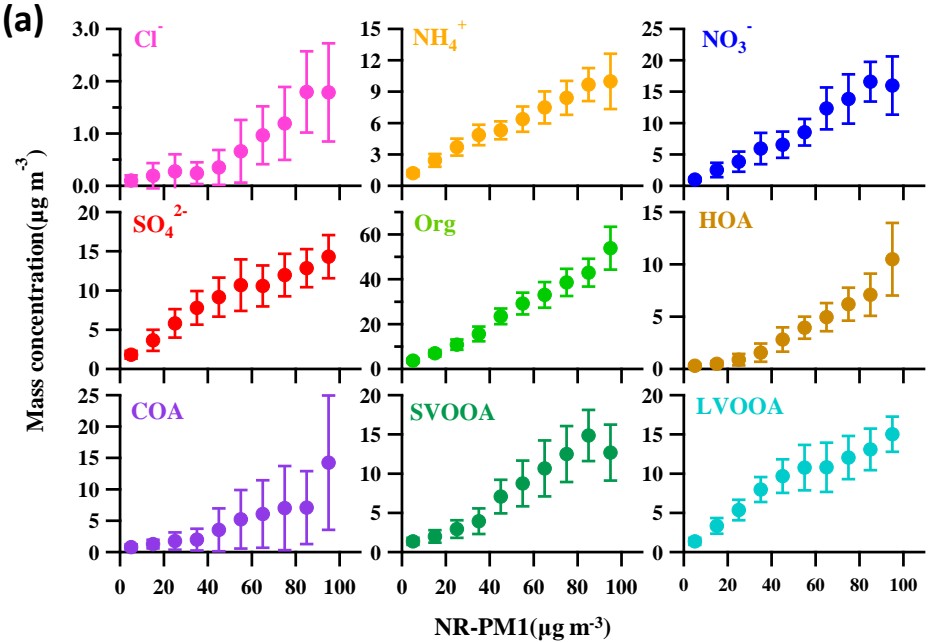

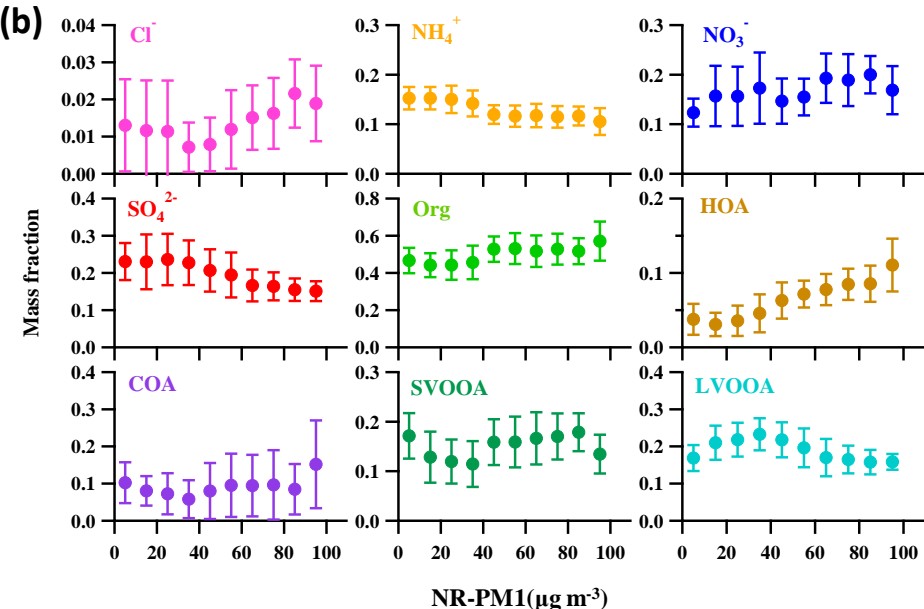

Fig. 7.





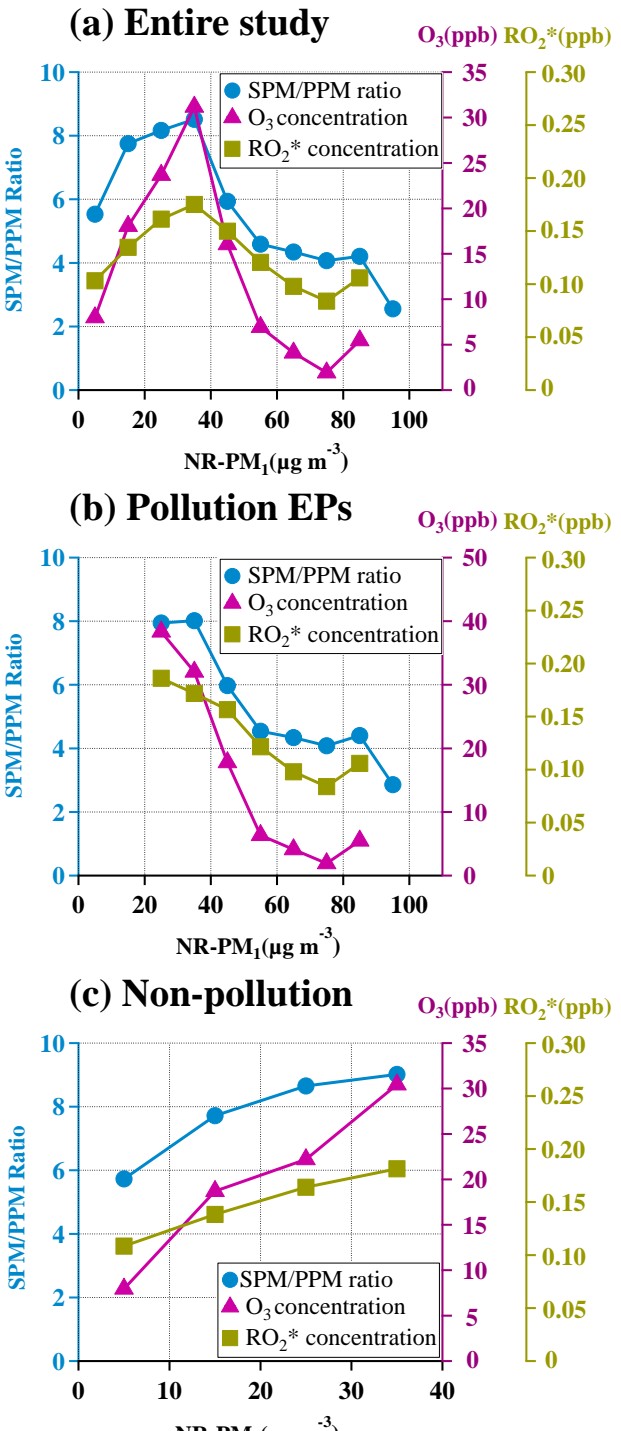

**Fig. 8.**





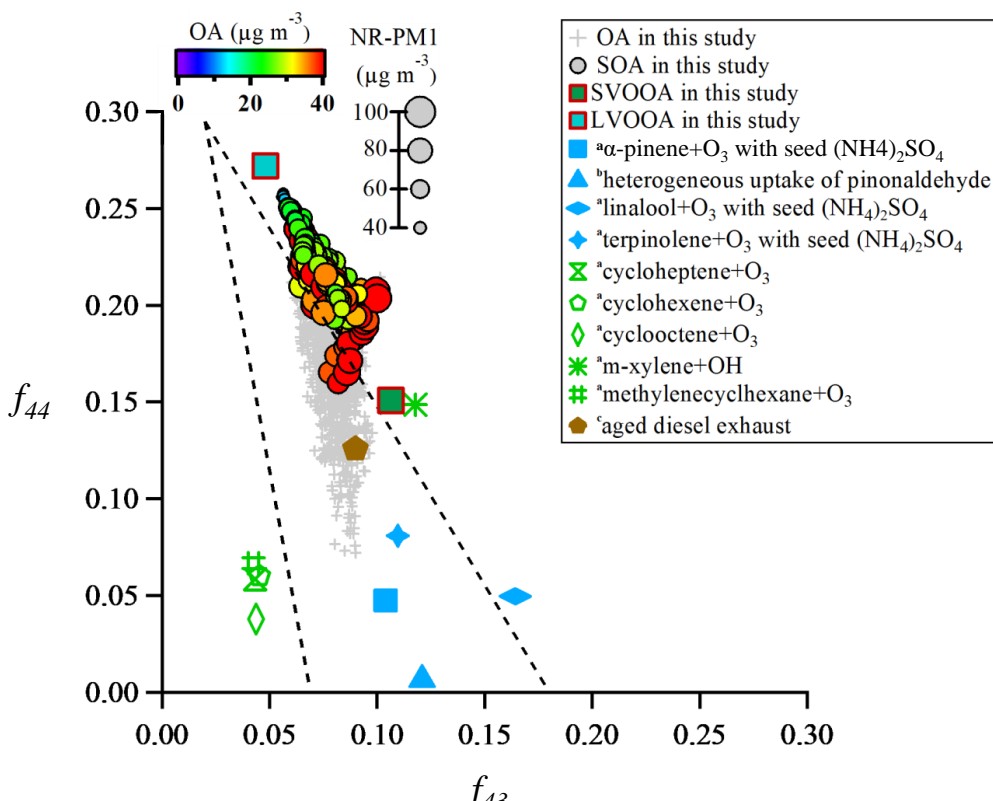

**Fig. 9.**


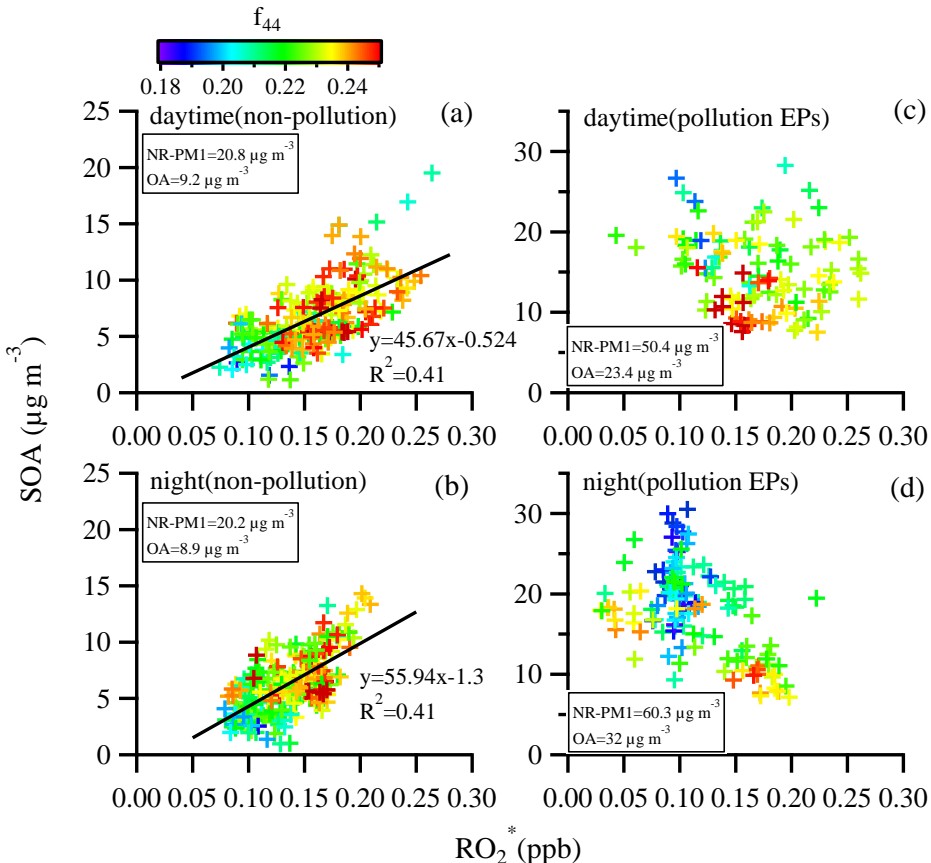

**Fig. 10.**





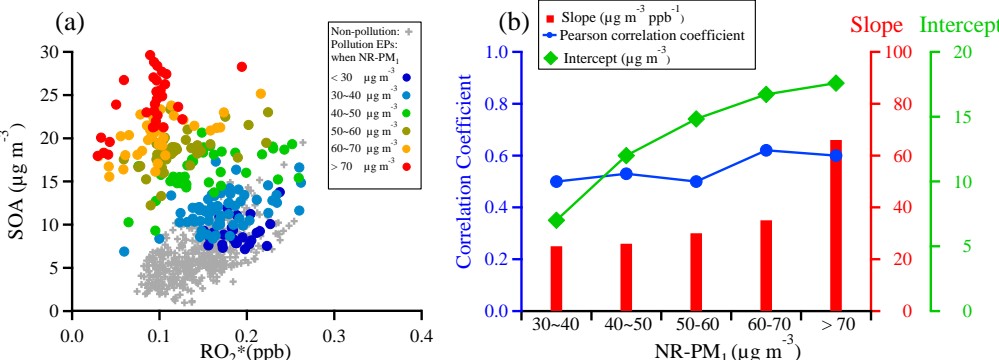

**Fig. 11.**



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
