# Peer review of "Characterization of submicron particles by Time-of-Flight Aerosol Chemical Speciation Monitor (ToF-ACSM) during wintertime: aerosol composition, sources and chemical processes in Guangzhou, China"

_Atmospheric Chemistry and Physics, 2019_

## Referee Comment (RC1) · Anonymous Referee #2 · 26 Feb 2020

This manuscript reports a ToF-ACSM measurement study of sub-micron particles conducted during winter time in Guangzhou, South China. PMF with ME-2 algorithm was applied on the dataset to identify the major sources of organic aerosols (OA). Discussions are made on concentrations, compositions, and sources ambient PM1, highlighting the important roles of SOA. Additionally, the relationship with SOA and peroxy radicals was examined to reveal the different mechanisms responsible for SOA formation between non-pollution period and pollution Eps. The manuscript is well written and

provides some interesting results for understanding ambient primary and secondary organic aerosol sources and processes. I would recommend the publication of this manuscript in Atmospheric Chemistry and Physics after the authors address the following comments.

1. The resolution and font sizes need to be improved? (e.g., Fig. 1, Fig. 4b and Fig. 5) 2. Please check the subscript in the texts and figures. 3. Page 7, Line 191-194: It would be good to add a more accurate discussion of the calculated composition-dependent CE values (e.g., range, highest frequency and uncertainty of CE values). 4. Page 8, Line 237-241: Recently, a large number of AMS/ACSM studies have been conducted in China in recent years. Is it possible to add more references and discuss with more results? 5. Page 15, Line 455-477: I suggest the authors put these parts in introduction and highlight the differences of your results from previous ones. 6. Page 28, Fig. 4: The factors of HOA and COA were resolved using the constrain mode (a-value), but SVOOA and LVOOA were identified using the PMF free mode. So, to be more directly clear for readers, the authors may consider adding the corresponding label in each mass spectrum of POA factors (e.g., constrained or a specific a-value) and SOA factors (e.g., unconstrained or free). 7. Page 29, Fig. 5: The diurnal profile of NOx appears to be bi-modal, yet no morning traffic feature is visible in the HOA diurnal plot during pollution Eps. Have the authors looked for the variation of HOA mass fraction during early rush hour? More explanation about the diurnal profile of HOA would be good. 8. Section 3.5.2: It would be interesting to see how the SOA changes during different conditions. The authors may consider adding the correlations between SVOOA and LVOOA with RO2*, perhaps in the supplement.

---

## Referee Comment (RC2) · Anonymous Referee #1 · 8 Mar 2020

This manuscript investigated the aerosol composition and sources in Guangzhou, China, with a focus on ACSM measurements and subsequent PMF analysis. Routine analysis and plots were made. The conclusions are solid, but not exciting. However, latter part of the manuscript brought advance to the knowledge base by investigating the SOA sources and formation mechanisms from the perspective of RO2 chemistry. It is found that SOA has moderate correlation with RO2* in non-pollution days, but not in polluted episodes. Some conclusions are inferred by this analysis and I will discuss

more later in the comment. This analysis distinguishes this manuscript from previous studies. I would like note that this type of novel analysis is missing from most of previous studies in China that are based on AMS+PMF analysis. Overall, I recommend publication with major revisions.

Major Comment Even though the analysis on the relationship between SOA and RO2 is novel, some discussions can be improved and I hope the authors will consider the following comments to make the manuscript hopefully more impactful. The major issue in this analysis is that SOA and RO2 have dramatically different lifetime (days vs seconds). From this point of view, I find the moderate correlation between SOA and RO2 in figure 10a and b intriguing, but hard to explain. My hypothesis for this correlation is that both SOA and RO2 are controlled by the amount of oxidants available. Another hypothesis is that they have similar diurnal variation (as shown in figure 5). It would be helpful to de-trend the SOA and RO2 and then make the correlation analysis. For example, correlating SOA and RO2 for data points at the same hour of day. In addition, the lifetime issue is especially important in polluted days when the air is more stagnant, SOA lingers for a long time, but RO2 lifetime is short due to enhanced NOx concentration. This is likely the main reason for the lack of correlation between SOA and RO2 in polluted days (i.e., figure 10c and d, Page 17 Line 495). Overall, the different lifetimes of SOA and RO2 should be kept in mind when interpreting any results. Correlation between RO2 and O3. Assume RO2 is at steady-state, then we can approximate d[RO2]/dt=k[O3][VOC]-k[RO2][NO]=0 [RO2]âĹİ ([O3][VOC])/([NO]) If the [VOC]/[NO] ratio doesn't change much, a correlation between RO2 and O3 may be expected. The equation here is over-simplified, but my point is that many discussions may start from or be explained by this type of simple mathematical derivation. In figure S4, the correlation between RO2 and O3 for the whole campaign is shown. I suggest the authors also categorize all data points into sub-groups (non-pollution vs polluted, day vs night, as figure 10) and show correlation relationship. The reason I suggest this is that based on figure 10c, polluted days only account for a small fraction of all data points. Thus, the relationship between RO2 and O3 for polluted days are

not clearly shown in the assembly of all data points. Figure 11 and Page 17 Line 505. By eyeballing, the correlation between RO2 and SOA each sub-group is very week. The p-value for linear regression must be included. Page 2 Line 62 and Page 17 Line 518. It is not clear why the intercept represents the extent of other SOA formation mechanisms. I think the related conclusions are overblown. Page 17 Line 513: RO2 concentration can not represent the amount of gas phase oxidation products. Figure 8. Please include all data points in this plot, in addition to the binned-average. Figure 10 and 11 and S4. Please use orthogonal linear regression, which considers the measurement uncertainty in both x- and y-axis.

Minor Comments: Page 4 Line 102: I believe the authors mean "inevitable", instead of "evitable". Page 6 Line 175: Even though the measurement details of RO2* have been described in previous studies, it is still beneficial to briefly mention how the measurements were done. Page 9 Line 254: The selection of a-value should be justified. Page 9 Line 274: Even in non-pollution period, rush hour peaks in HOA are expected. Thus, the lack of diurnal variation in HOA in this study is alarming. It may be due to the low resolution of ACSM. Page 10 Line 307: Replace "high volatility" with "semi-volatile". Page 16 Line 490: the trend between f44 and SOA concentration is not clear in figure 10. Regarding the claimed conclusion that f44 of SOA decreases with increase of SOA concentration, I suggest the authors to check if there is any "contamination" in PMF analysis. In other words, in polluted days, is some POA apportioned into SOA by PMF analysis? This can be done by either run PMF on non-pollution and polluted days separately and see if the fraction of SOA changed, or check the residual of POA characteristic ions during polluted days. The former approach is more reliable.
* * *

---

## Author Comment (AC1) · 4 May 2020

**Response to the referee's comments**

We would like to thank the reviewer for valuable comments and suggestions. We have addressed all raised issues in the revision accordingly. Please kindly find our following point-by-point responses. The reviewer's comments in black and our responses in blue. Any amendments in the revised manuscript are highlighted in red.

*Response to Reviewer #2*

This manuscript reports a ToF-ACSM measurement study of sub-micron particles conducted during winter time in Guangzhou, South China. PMF with ME-2 algorithm was applied on the dataset to identify the major sources of organic aerosols (OA). Discussions are made on concentrations, compositions, and sources ambient PM1, highlighting the important roles of SOA. Additionally, the relationship with SOA and peroxy radicals was examined to reveal the different mechanisms responsible for SOA formation between non-pollution period and pollution Eps. The manuscript is well written and provides some interesting results for understanding ambient primary and secondary organic aerosol sources and processes. I would recommend the publication of this manuscript in Atmospheric Chemistry and Physics after the authors address the following comments.

*[A]:* Thank you for the comments and valuable suggestions. We have changed accordingly. Please find our point-by-point responses below.

**Comments:**

**1.** The resolution and font sizes need to be improved? (e.g., Fig. 1, Fig. 4b and Fig. 5).

*[A]:* We have increased the resolution and font sizes of the figures.

**2.** Please check the subscript in the texts and figures.

*[A]:* We have doubly checked the entire manuscript to ensure no typos with the subscript.

**3.** Page 7, Line 191-194: It would be good to add a more accurate discussion of the calculated composition dependent CE values (e.g., range, highest frequency and uncertainty of CE values).

*[A]:* We have added several sentences to discuss the calculated composition dependent CE values in the revision (lines 217-220, pages 7-8):

"The results showed that only about 1% of samples (78 of 6623) had CE values larger than 0.45 (others are 0.45), with the largest value being 0.578. Hence the influence induced by fluctuation of the CE values is negligible and we chose a CE value of 0.45 for the ACSM measurements in this study."

**4.** Page 8, Line 237-241: Recently, a large number of AMS/ACSM studies have been conducted in China in recent years. Is it possible to add more references and discuss with more results?

*[A]:* We thank the reviewer for updating us on this information. We have included 11 additional publications, with a comprehensive summary being added in the revision (Table 1, in the revised manuscript or below). In addition, we have also modified in Figure 3 by including more measurement data for Beijing and Lanzhou in the revision.

Here we only compare measurements during winter season, corresponding to our measurement periods. Our survey shows that SOA formation is significantly influenced under different underlying surfaces (urban, suburban, and country). Nevertheless, additional survey adds more measurement data into the NR-PM$_1$ pool. However, our original conclusion of increasing the fraction of SOA in OA from north to south still holds. We have modified the paragraph that describes NR-PM$_1$ measurements in China in the revision (line 267-269, page 9). "Furthermore, Table 1 shows that the SOA fraction is generally enhanced from winter to summer for a specific site in China. In addition, Table 1 also revealed that SOA formation is significantly influenced under different underlying surfaces (urban, suburban, and country)."

**5.** Page 15, Line 455-477: I suggest the authors put these parts in introduction and highlight the differences of your results from previous ones.

*[A]:* We thank the reviewer for valuable suggestions. We have made some modifications and moved these sentences to introduction section in the revision (lines 140-159, page 5).

In addition, we have added one sentence to highlight differences of our results from previous studies in the revision (lines 177-178, page 6).

"Possible mechanisms for wintertime SOA formation were explored through introducing RO$_2$$^*$ as a proxy for gas-phase oxidation capacity during both daytime and nighttime."

**6.** Page 28, Fig. 4: The factors of HOA and COA were resolved using the constrain mode (a-value), but SVOOA and LVOOA were identified using the PMF free mode. So, to be more directly clear for readers,

the authors may consider adding the corresponding label in each mass spectrum of POA factors (e.g., constrained or a specific a-value) and SOA factors (e.g., unconstrained or free).

*[A]:* According to the reviewer's suggestions, we have added the corresponding labels in Figure 4.

[Figure]

**Figure 4.** The mass spectra and time series of the four OA components (HOA, COA, SVOOA, and LVOOA).

**7.** Page 29, Fig. 5: The diurnal profile of NOx appears to be bi-modal, yet no morning traffic feature is visible in the HOA diurnal plot during pollution Eps. Have the authors looked for the variation of HOA mass fraction during early rush hour? More explanation about the diurnal profile of HOA would be good.

*[A]:* We thank the reviewer for pointing this out. Though it is small, there was a small HOA peak at about 8:00 in Fig. 5 in the manuscript. In fact, strong traffic emissions from heavy duty vehicles during midnight to 6:00 in the early morning weaken the morning rush hour peak. Similar features were frequently reported in previous studies (Sun et al., 2013; Qin et al., 2017; Huang et al., 2019). The rapidly rising boundary layer after 7:00 would be possibly another reason for diluting PM accumulation from rush hour traffic. Besides, if we discuss the variation of HOA mass fraction, a clear morning peak was observed (Fig. S12, see it in supplementary or below). Thus, the diurnal feature of HOA during pollution Eps is reasonable. We have added a sentence in the revision for clarification (lines 298-299, page 10): "Besides, effects of emissions from heavy duty vehicles and the rapidly rising boundary layer after 7:00 would also account for the insignificant peak of HOA during morning rush hour."

[Figure]

**Figure S12.** The diurnal variations of 4 OA components mass fractions for pollution EPs.

**8.** Section 3.5.2: It would be interesting to see how the SOA changes during different conditions. The authors may consider adding the correlations between SVOOA and LVOOA with $RO_2^*$, perhaps in the supplement.

*[A]:* We have plotted dependence of SVOOA/LVOOA concentrations on $RO_2^*$ concentration during non-pollution periods and pollution periods in Figure S15 (see it in supplementary or below). It is shown that better correlations between LVOOA and $RO_2^*$ than between SVOOA and $RO_2^*$. In addition, the slope from LVOOA vs $RO_2^*$ is higher than that from SVOOA vs $RO_2^*$, implying transformation of SVOOA to LVOOA. In contrast, neither LVOOA nor SVOOA were well correlated to $RO_2^*$, possibly due to strong heterogeneous/multiphase reactions during pollution EPs as discussed in the text.

We have added several sentences in the revision to reflect the correlations between SVOOA/LVOOA concentrations and RO$_2$* concentration (lines 491-495, page 17).

"In addition, correlations between SVOOA/LVOOA and RO$_2$ were explored by plotting dependence of SVOOA/LVOOA concentrations on RO$_2$* concentration during non-pollution periods and pollution periods (Fig. S15). The results show that better correlations and larger slope for LVOOA vs RO$_2$* than for SVOOA vs RO$_2$* during non-pollution periods. In contrast, neither LVOOA nor SVOOA were correlated to RO$_2$* during pollution EPs."

[Figure]

**Figure S15.** Scatter plots between RO$_2$* and SVOOA/LVOOA for non-pollution periods and pollution EPs.

**Table 1.** Summary of reported NR-PM$_1$ measurements in China.

[revised manuscript text omitted]

---

## Author Comment (AC2) · 4 May 2020

We would like to thank the reviewer for valuable comments and suggestions. We have addressed all raised issues in the revision accordingly. Please kindly find our following point-by-point responses. The reviewer's comments in black and our responses in blue. Any amendments in the revised manuscript are highlighted in red.

*Response to Reviewer #1*:

This manuscript investigated the aerosol composition and sources in Guangzhou, China, with a focus on ACSM measurements and subsequent PMF analysis. Routine analysis and plots were made. The conclusions are solid, but not exciting. However, latter part of the manuscript brought advance to the knowledge base by investigating the SOA sources and formation mechanisms from the perspective of RO2 chemistry. It is found that SOA has moderate correlation with RO2* in non-pollution days, but not in polluted episodes. Some conclusions are inferred by this analysis and I will discuss more later in the comment. This analysis distinguishes this manuscript from previous studies. I would like note that this type of novel analysis is missing from most of previous studies in China that are based on AMS+PMF analysis. Overall, I recommend publication with major revisions.

*[A]*: We would like to thank the reviewer for valuable comments and suggestions. Please find below our point-by-point responses.

**Comments:**

**1.** Even though the analysis on the relationship between SOA and $RO_2$ is novel, some discussions can be improved and I hope the authors will consider the following comments to make the manuscript hopefully more impactful. The major issue in this analysis is that SOA and $RO_2$ have dramatically different lifetime (days vs seconds). From this point of view, I find the moderate correlation between SOA and $RO_2$ in figure 10a and b intriguing, but hard to explain. My hypothesis for this correlation is that both SOA and $RO_2$ are controlled by the amount of oxidants available. Another hypothesis is that they have similar diurnal variation (as shown in figure 5). It would be helpful to de-trend the SOA and $RO_2$ and then make the correlation analysis. For example, correlating SOA and $RO_2$ for data points at the same hour of day. In addition, the lifetime issue is especially important in polluted days when the air is more stagnant, SOA lingers for a long time, but $RO_2$ lifetime is short due to enhanced NOx concentration. This is likely the

main reason for the lack of correlation between SOA and $RO_2$ in polluted days (i.e., figure 10c and d, Page 17 Line 495). Overall, the different lifetimes of SOA and $RO_2$ should be kept in mind when interpreting any results.

*[A]:* We thank the reviewer for raising the important points on the correlations between SOA and $RO_2$*. We agree with the reviewer that lifetimes between SOA and $RO_2^*$ are dramatically different. However, following the reviewer's suggestions, we further analyze the measurement data and find that the lifetime effects probably play a minor role in poor correlations between SOA and $RO_2^*$ during pollution periods. Our results further confirm that gas-phase oxidation and gas-phase oxidation in combination with heterogenous reactions are respectively the main reasons for moderate correlations between SOA and $RO_2^*$ during non-pollution periods and poor correlations during pollution periods.

(1) We believe that moderate correlations between SOA and $RO_2^*$ in non-pollution period are due to gas-phase oxidation of VOCs that leads to formation of both $RO_2^*$ and SOA (Details can be found in 147-159 in the revision). Thus, excluding heterogeneous reactions, intense gas phase oxidation (not only related to the amount of oxidants but VOCs) is expected to enhance the whole system of $RO_2$*+SOA. Otherwise, both $RO_2$* and SOA concentrations will decrease under weak gas phase oxidation. This is where the correlation between SOA and $RO_2$* comes from.

According to the reviewer's suggestions, for the non-pollution period, we de-trended the correlations between the two species by plotting the dependence of SOA on $RO_2$* concentration for the same short period (every 2 hours, to ensure enough points) of every day during non-pollution period in Fig. S13 (see it in supplementary or below). Like Figs. 10a and b in the manuscript, moderate correlations are seen from the de-trending plots. The results suggest the relationship between SOA and $RO_2^*$, especially the upward trend of SOA with increasing $RO_2^*$, is convincing. Thus, similar diurnal variations between SOA and $RO_2^*$ in non-pollution period are the result of shared influence from gas phase oxidation of VOCs.

[Figure]

**Figure S13.** Dependence of SOA on $RO_2^*$ concentration for the same short period of every day (i.e., every 2 hours) during non-pollution periods. Note that all the correlations are statistically significant (p-value < 0.01).

**(2)** During pollution periods, however, the correlations between SOA and $RO_2^*$ become a little more complicated than those during non-pollution periods. We agree with the reviewer that lifetimes between SOA and $RO_2^*$ are dramatically different (days vs seconds). However, here we show that the dramatic differences of lifetimes between the two species are not likely the main reasons which lead to poor correlations between them. As we mentioned in the paper, the odd oxygen ($O_x = O_3 + NO_2$) can be used as a robust photochemical indicator, which has a lifetime of about one day, a similar magnitude to that of SOA (Schaub et al., 2007; Lamsal et al., 2010; Valin et al., 2013). Here we present daytime relationship of $O_x$ with SOA and $RO_2^*$ during non-pollution and pollution EPs respectively (Fig. S14, see it in supplementary or below). Reasonably good correlations were found between $RO_2^*$ and $O_x$ during both non-pollution and pollution EPs. However, only reasonably good correlation between SOA and $O_x$ was found during non-pollution periods, while no correlation at all was found during pollution EPs, neither

was found for the correlation between SOA and $RO_2^*$ during pollution EPs (Fig. 10c in the revision). If the dramatic difference of lifetime was to play an important role in the correlation as the reviewer hypothesizes, the reasonably good correlation between $RO_2^*$ and $O_x$ in non-pollution period shouldn't exist during pollution EPs, just like what happen to $RO_2^*$ and SOA. Besides, when we discuss the relationship between SOA and $O_x$ (similar life time), the same pattern as that for SOA and $RO_2^*$ (Fig. S15 c and d) is revealed. Hence, we believe that the lifetime effects probably play an insignificant role in the poor correlation between SOA and $RO_2^*$ during pollution EPs. We think that no correlations between SOA and $RO_2^*/Ox$ during pollution EPs implied dramatical changing of SOA formation mechanisms.

As the reviewer pointed out, the enhanced $NO_x$ during polluted days would influence the lifetime of $RO_2^*$. As $NO_x$ concentration increases, more $RO_2^*$ was consumed. Then the correlation between $RO_2^*$ and $O_x$ (also SOA) may become worse with significantly enhanced $NO_x$ levels. However, this influence is probably not significant in this study because a good correlation was still found between $RO_2^*$ and $O_x$ during pollution EPs (Fig. S15b).

[Figure]

**Figure S14.** Daytime dependence of $O_x$ on $RO_2^*$ (a, b) and dependence of SOA on $O_x$ (c, d) for pollution EPs and non-pollution period.

[revised manuscript text omitted]

**2.** Correlation between $RO_2$ and $O_3$. Assume $RO_2$ is at steady-state, then we can approximate $d[RO2]/dt=k[O3][VOC]-k[RO2][NO]=0$ $[RO2]≈ ([O_3][VOC])/([NO])$ If the [VOC]/[NO] ratio doesn't change much, a correlation between $RO_2$ and $O_3$ may be expected. The equation here is over-simplified, but my point is that many discussions may start from or be explained by this type of simple mathematical derivation.

*[A]:* We thank the reviewer for valuable suggestions. According to recent measurements in Guangzhou, about 67%, 17% and 16% of VOCs are alkanes, alkenes and aromatic hydrocarbons respectively (Zou et al., 2015). Hence during daytime, it is likely that the predominant sources of $RO_2*$ are from OH-initiated hydrocarbon oxidation, complicated by a probably minor contribution from ozone- alkene/aromatic hydrocarbon oxidation. In fact, based on previous VOCs (Zou et al., 2015) and OH (Rohrer et al., 2014) measurement data in Guangzhou, along with $O_3$ concentration in this study and rate constants reported in Atkinson et al. (2003), the OH contribution to $RO_2*$ is estimated 100 times larger than $O_3$ contribution. During nighttime, however, since it is likely that the predominant sources of $RO_2*$ are from $NO_3$-initiated hydrocarbon oxidation, along with a minor contribution from ozone-alkene/aromatic hydrocarbon oxidation (Volkamer et al., 2010; Stone et al., 2014). In either case, the relationship between $RO_2*$ and ozone is not straightforward. Hence it is unlikely to connect them with a simple and explicit mathematical form.

In fact, in this paper, we employ $O_x$ ($O_x=O_3+NO_2$) instead of ozone itself as a proxy when discussing the effects of photochemistry on SOA formation (Herndon et al., 2008). Here $NO_2$ is used to compensate $O_3$ consumption. Here we plot $O_x$ concentration as a function of $RO_2^*$ concentration for different scenarios (non-pollution daytime, non-pollution nighttime, pollution daytime and pollution night time) (Fig. S4 a-d, see it in supplementary or below). The results show that good correlations between $O_x$ and $RO_2^*$ were only seen during daytime when photochemistry occurs while poor or no correlations at all were observed during nighttime when photochemistry is shut off. Interestingly, when we plot ozone concentration as a function of $RO_2^*$ concentration for the four scenarios, good correlations between ozone and $RO_2^*$ were all seen during regardless of daytime or nighttime (Fig. S4 e-h). As mentioned above, good correlation between $RO_2^*$ and $O_3$ should come from shared photochemistry during daytime. In contrast, $RO_2^*$ is mainly contributed by $NO_3/O_3$-initiated VOCs oxidation during nighttime. Previous studies have proved that nocturnal $NO_3$ radicals mainly comes from $NO_2$ oxidation by $O_3$, i.e., $NO_2+O_3=NO_3+O_2$ (Volkamer et al., 2010; Stone et al., 2014). Thus, $RO_2^*$ is always related to $O_3$ concentration at night, likely explaining their good nocturnal correlations.

[Figure]

**Figure S4.** Dependence of $O_x$ and $O_3$ on $RO_2$* for different scenarios (non-pollution daytime period, pollution daytime EPs, non-pollution nighttime period, and pollution nighttime EPs). All the regressions are orthogonally linear.

**3.** In figure S4, the correlation between $RO_2$ and $O_3$ for the whole campaign is shown. I suggest the authors also categorize all data points into sub-groups (non-pollution vs polluted, day vs night, as figure 10) and show correlation relationship. The reason I suggest this is that based on figure 10c, polluted days only

account for a small fraction of all data points. Thus, the relationship between RO2 and O3 for polluted days are not clearly shown in the assembly of all data points.

*[A]:* According to the reviewer's suggestions, we have adjusted Figure S4 and showed the correlation between $RO_2$ and $O_3$ with subgroups. Please refer to answer #2.

**4.** Figure 11 and Page 17 Line 505. By eyeballing, the correlation between RO2 and SOA at each sub-group is very week. The p-value for linear regression must be included.

*[A]:* We included the p-values into Figure 11 in the original version (now Figure 12 in the revision).We also added a sentence to the caption of the figure, "all correlations of the concentration segments were statistically significant (p-value < 0.01, see detailed statistical information in Table S3)." Additionally, a table (Table S3) with detailed statistic information was added into the supplementary.

We agree with the reviewer that the correlations between SOA and $RO_2$* during pollution periods are not as strong as those during non-pollution periods probably due to fewer points. Nevertheless, we show that a more viable dependence trend of SOA on $RO_2$* in pollution EPs can be seen by dividing samples into 6 sub-groups based on different $PM_1$ mass intervals (Figure 12).

**Table S3.** P-value, T-value, number of points (n), and Pearson Correlations between SOA and $RO_2$* for different $NR\text{-}PM_1$ concentration intervals.

| $NR\text{-}PM_1$ ($\mu g\ m^{-3}$) | r | n | T | p |
|---|---|---|---|---|
| < 30 | 0.31 | 47 | 5.07 | <0.001 |
| 30-40 | 0.50 | 60 | 4.40 | <0.001 |
| 40-50 | 0.53 | 35 | 3.56 | 0.0012 |
| 50-60 | 0.49 | 41 | 3.51 | 0.0011 |
| 60-70 | 0.62 | 40 | 4.93 | <0.001 |
| >70 | 0.59 | 30 | 3.85 | <0.001 |

[Figure]

**Figure 12.** (a) Scatter plots of SOA and RO$_2$* during pollution EPs at different NR-PM$_1$ mass concentration segments, and (b) Correlation coefficients, slopes, and intercepts of linear regressions between SOA and RO$_2$* for NR-PM$_1$ mass concentration segments ranging from 30-40 to > 70 μg m$^{-3}$. The regressions are orthogonally linear, and all correlations of the concentration segments were statistically significant (p-value < 0.01, see detailed statistical information in Table S3).

**5.** Page 2 Line 62 and Page 17 Line 518. It is not clear why the intercept represents the extent of other SOA formation mechanisms. I think the related conclusions are overblown.

*[A]:* We thank the reviewer for pointing this out. As we show in answer #1 that that gas-phase oxidation and gas-phase oxidation in combination with heterogenous reactions are respectively the main reasons for moderate correlations between SOA and RO$_2$* during non-pollution periods and poor correlations during pollution periods. As also mentioned in the above answer, although the correlations between SOA and RO$_2$* during pollution periods are not as strong as those during non-pollution periods, the extrapolation of these regressions, that is, the intercepts might imply mechanisms of SOA formation as more SOA is formed with increasing PM concentrations. In addition, the intercept from the regressions between SOA and RO$_2$* during non-pollution periods is essentially close to zero, leading us to believe that the intercepts likely correspond to the amount of SOA formed from other mechanisms (i.e., heterogenous reactions) other than gas-phase oxidation.

6.Page 17 Line 513: RO$_2$ concentration can not represent the amount of gas phase oxidation products

*[A]:* We agree with the reviewer and have removed the relevant sentence.

**7.** Figure 8. Please include all data points in this plot, in addition to the binned-average.

*[A]:* We thank the reviewer for the suggestion. We have modified Figure 8 in revision by including all data points and separating it into several panels. We have hence changed the caption of Figure 8 as shown below.

[Figure]

**Figure 8.** Dependences of SPM/PPM ratio, concentrations of the atmospheric oxidants ($O_3$ and $RO_2^*$) on NR-$PM_1$ mass loading for (a) entire study, (b) non-pollution period and (c) pollution EPs. The binned data are also presented as solid circles with an interval of 10 µg m$^{-3}$ NR-$PM_1$. The error bars are standard deviations.

**8.** Figure 10 and 11 and S4. Please use orthogonal linear regression, which considers the measurement uncertainty in both x- and y-axis.

*[A]:* Per the reviewer's suggestions, we have replaced Figure 10, Figure 11 (now Figure 12 in the revision), and Figure S4 using orthogonal linear regressions (Minitab 2019, www.minitab.com). The renewed figures don't change our original conclusions. We also mentioned this in captions of these figures.

**Minor Comments**

**1.** Page 4 Line 102: I believe the authors mean "inevitable", instead of "evitable".

*[A]:* Corrected (line 103, page 4).

**2.** Page 6 Line 175: Even though the measurement details of RO2* have been described in previous studies, it is still beneficial to briefly mention how the measurements were done.

*[A]:* Agree, we have added several sentences to describe the measurement details of $RO_2*$ in the revision (lines 194-200, page 7).

"The concentrations of the total peroxy radicals ($RO_2* = \Sigma RO_2^{\cdot} + HO_2$) were measured with a dual-channel PERCA (Peroxy Radical Chemical Amplification) instrument (Yang et al., 2018; Yang et al., 2019). In this instrument, ambient mixing ratios of $RO_2*$ radicals were converted to a larger amount of $NO_2$ by reacting with NO and CO. The amplified $NO_2$ concentrations were then measured with a portable broadband cavity enhanced spectrometer (BBCES) with a precision of 40 pptv ($1\sigma$, with 21 s data acquisition time) (Fang, et al., 2017). The total uncertainty of the PERCA instrument was about 10% with a precision of about 0.4 pptv ($1\sigma$, 21 s)."

**3.** Page 9 Line 254: The selection of a-value should be justified.

*[A]:* We thank the reviewer for pointing this out. We have now added several sentences in supplementary in the method section, "The results showed that an unreasonably high proportion of m/z 44 were presented in COA profiles for solutions with a-values of 0.5 and 0.7. We hence adopt 4 factors and an a-value of 0.3 as the optimal solution. The results from ME-2 are shown in Figures S5-S11".

In addition, we have also added one sentence in the revision (lines 278-279, page 9), "Detailed selection of a-value can be found in supplementary (method section)."

**4.** Page 9 Line 274: Even in non-pollution period, rush hour peaks in HOA are expected. Thus, the lack of diurnal variation in HOA in this study is alarming. It may be due to the low resolution of ACSM.

*[A]:* We agree with the reviewer that the resolution of ACSM is low compared to regular AMS. The HOA concentration during non-pollution periods after ME-2 factorization is low ($< 1$ $\mu g$ $m^{-3}$), close to an estimated detection limit of 0.7 $\mu g$ $m^{-3}$ for OA, leading to lack of diurnal variations in HOA. However, it is unlikely to affect other organic factors since the concentrations of other organic components are much higher than the estimated detection limit for OA.

For clarification, we have added several sentences in the revision (lines 299-302, page 10), "In comparison, the HOA concentration showed almost no variations during non-pollution period (even

during rush hours), which likely arose from its extremely low value ($< 1$ µg m$^{-3}$ which was close to an estimated detection limit of 0.7 µg m$^{-3}$ for OA with the ToF-ACSM)."

**5.** Page 10 Line 307: Replace "high volatility" with "semi-volatile".

*[A]:* We have corrected it (line 333, page 11).

**6.** Page 16 Line 490: the trend between f44 and SOA concentration is not clear in figure 10. Regarding the claimed conclusion that f44 of SOA decreases with increase of SOA concentration, I suggest the authors to check if there is any "contamination" in PMF analysis. In other words, in polluted days, is some POA apportioned into SOA by PMF analysis? This can be done by either run PMF on non-pollution and polluted days separately and see if the fraction of SOA changed, or check the residual of POA characteristic ions during polluted days. The former approach is more reliable.

*[A]:* According to the reviewer's suggestions, we plotted the dependence of SOA concentration on f44 for different scenarios (non-pollution daytime, non-pollution nighttime, pollution daytime and pollution night time) as is shown in Fig. S16 (see it in supplementary or below). The trends between SOA and f44 are clear: SOA concentrations increase with increasing f44 during non-pollution periods while opposite trends are observed during pollution periods. In addition, we ran Me-2 separately for non-pollution periods and pollution EPs (Fig. S17, see it in supplementary or below). The results from those additional runs show similar fractions and concentrations of SOA, confirming reliability of our original results.

[Figure]

**Figure S16.** Scatter plots between SOA and f44 for different scenarios (non-pollution daytime, non-pollution nighttime, pollution daytime and pollution night time).

[Figure]

**Figure S17.** Comparison between combined ME-2 and separate ME-2, along with their correlations.